# Arsenic Contamination of Groundwater Is Determined by Complex Interactions between Various Chemical and Biological Processes

Zahid Hassan [1,2] and Hans V. Westerhoff [1,3,4,5,*]

1  Department of Molecular Cell Biology, A-Life, Faculty of Science, Vrije Universiteit Amsterdam, 1081 HV Amsterdam, The Netherlands; zahid@geb.jnu.ac.bd
2  Department of Genetic Engineering and Biotechnology, Jagannath University, Dhaka 1100, Bangladesh
3  School of Biological Sciences, Faculty of Biology, Medicine and Health, The University of Manchester, Manchester M13 9PT, UK
4  Synthetic Systems Biology and Nuclear Organization, Swammerdam Institute for Life Sciences, University of Amsterdam, 1098 XH Amsterdam, The Netherlands
5  Stellenbosch Institute of Advanced Studies (STIAS), Wallenberg Research Centre at Stellenbosch University, Stellenbosch 7600, South Africa
*  Correspondence: hvwesterhoff@gmail.com

**Abstract:** At a great many locations worldwide, the safety of drinking water is not assured due to pollution with arsenic. Arsenic toxicity is a matter of both systems chemistry and systems biology: it is determined by complex and intertwined networks of chemical reactions in the inanimate environment, in microbes in that environment, and in the human body. We here review what is known about these networks and their interconnections. We then discuss how consideration of the systems aspects of arsenic levels in groundwater may open up new avenues towards the realization of safer drinking water. Along such avenues, both geochemical and microbiological conditions can optimize groundwater microbial ecology vis-à-vis reduced arsenic toxicity.

**Keywords:** systems biology; systems chemistry; arsenic toxicity; subsurface arsenic removal; arsenic microbial ecology; bioremediation; safe drinking water; iron in drinking water wells; bioaugmentation; systems toxicology





## 1. Introduction

Chemical pollution of groundwater poses a serious threat to public health (Figure S1). The pollution can either be due to industrial discharges and other anthropogenic activities or it can occur naturally. Arsenic is known as the "king of poison" or "hidden killer" [1]. The leaching of soil, weathering of rocks, and agricultural runoff all introduce arsenic into groundwater [2]. Natural sources further include seawater, arsenic-bearing minerals, volcanic emissions, and rivers originating in the Himalayas [3,4].

Inorganic arsenic is naturally present at high levels in groundwater in many countries across the globe, including the Americas (e.g., Argentina, Chile, Mexico, USA) [5], Asia (e.g., India, Pakistan, Bangladesh, Nepal, China, Taiwan), Southeast Asia (e.g., Indonesia, Thailand, Vietnam, Cambodia, Myanmar) [5–13], and Europe (e.g., Hungary, France, Germany, Romania, Italy) [14–18] (Figure 1). Nearly 200 million people worldwide are at risk of arsenic poisoning, including 180 million in Asia [19–21]. In nearly 108 countries, arsenic in groundwater exceeds the maximum of 10 µg/L recommended by the WHO. In total, 1 person out of every 60 people lives in a region where the concentration of arsenic in groundwater is 50 µg/L or above [22]. Bangladesh is the country that is worst off (Figure 1): groundwater is its major source of pathogen-free drinking water [23,24]. Ineffective water purification and sewage systems as well as periodic monsoons, cyclones, flooding, drought, and salinity complicate access to reliable drinking water. Approximately,

80 million inhabitants of Bangladesh are exposed to groundwater with concentrations above 50 μg/L and 35 million are potentially exposed to even higher concentrations of arsenic (50–300 μg/L) in drinking water [25,26]. The arsenic concentration in some tube wells is as high as 4.7 mg/L [27]. The WHO called this the largest mass poisoning of a population in history [28] and increased its guideline for maximal arsenic in drinking water in Bangladesh to 50 μg/L [29,30]. These numbers are extreme for this country, but many other countries are also troubled by arsenic in groundwater [8].

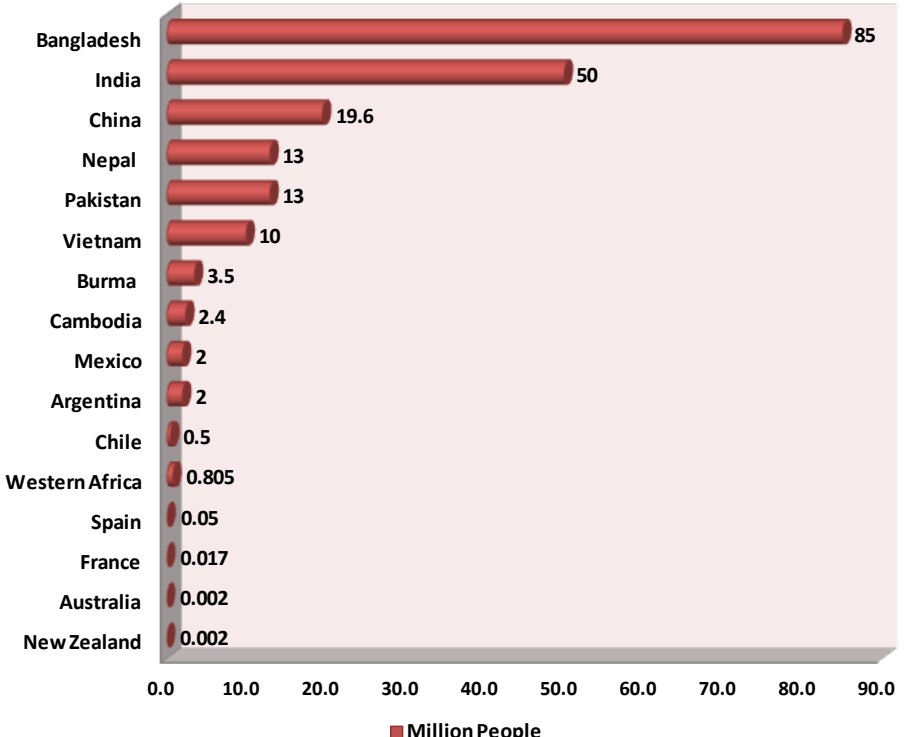

**Figure 1.** Estimated numbers of people (in millions) per country exposed to arsenic-contaminated groundwater (data were obtained from [19]).

The toxicity of arsenic in groundwater depends on a multitude of factors. These include chemical aspects like the conversion of the arsenic between its multiple chemical forms; the precipitation and subsequent adsorption of these forms with various metal oxides, particularly of manganese, iron, or aluminium [31]; the interaction of different arsenic species with various chemical forms of iron; the biochemical reactions of the arsenicals in the human; the interactions of arsenicals with metabolism; various anthropogenic activities that occur through mining and subsequent chemical modifications that occur during smelting for industrial, pharmaceutical, and agricultural uses; and various other groundwater characteristics such as ambient redox potential, pH, nitrate, organic carbon, sulfur, and many other chemical concentrations [15].

The various processes involved depend on each other nonlinearly, such as in the precipitation of arsenic which is dependent on the ambient redox potential or the precipitation of ferric but not ferrous iron, alone or together with arsenate (more than arsenite). These complex, nonlinear dependencies make this a case where integrative approaches become useful if not necessary. Systems biology and systems chemistry integrate concrete experimental data with existing physical, chemical, and biological knowledge and are often assisted by mathematical equations and modeling [32]. In Section 2 of this review, we discuss the systems chemistry of arsenic toxicity, with a particular emphasis on the interactions between the processes.

Toxicity further depends on the biochemistry in the various microbes present in the groundwater and surrounding soil, as well as on the growth rates of these organisms, which,

in turn, depend on but also influence factors such as pH, pO$_2$, nitrate levels, and organic carbon levels. The corresponding nonlinearities again require simultaneous evaluation. In Section 3, we discuss how this increases the level of complexity to the level of systems biology, presenting it in the context of systems chemistry. Also, this focus on systems biology embedded in systems chemistry is new compared to other recent reviews (e.g., [8], see Section 5).

After surveying both the systems chemistry (Section 2) and the systems biology (Section 3), Section 4 integrates the two. We discuss how the microbiology and chemistry may be manipulated such that the arsenic toxicity of groundwater used as drinking water is reduced. The interventions again have nonlinear effects. Some of these may have been responsible for the relative lack of success of subsurface arsenic removal approaches (SAR). We therefore discuss how consideration of systems biology aspects of SAR may lead to new avenues towards the realization of safer drinking water by creating conditions that optimize groundwater microbial ecology vis-à-vis reduced arsenic toxicity. We then sketch a *biotic* subsurface arsenic removal strategy (bSAR) that may well be worth for developing and which should heed the various nonlinear interactions in the system.

In recent years, an enormous amount of information has become available on both the chemistry and the microbiology of arsenic in groundwater. This includes the appearance of pangenome-wide sequence data, another aspect of systems biology. This has produced an avalanche of data against a fading background of general, e.g., thermodynamic, principles. All these data and principles need to be integrated in order to come to a functional understanding of the topic that may spur the (bio)remediation of arsenic toxicity. This paper is not only a review, but at the same time it initiates a discussion of how to bring this integration about; it takes a step towards a more integral approach to the arsenic toxicity of groundwater.

Because so many people on this planet are affected, knowingly or unknowingly, possibly to detrimental extents that are equally unappreciated (e.g., the effect of arsenic toxicity on human intelligence), we here sketch avenues that may enable the integration of information through systems biology and systems chemistry, with the ultimate goal of successful bioremediation.

## 2. Systems Chemistry

The term "systems chemistry" was first used in 2005 by Von Kiedrowski and colleagues [33] to describe the kinetic and computational analysis of a nearly exponential organic replicator. Later, systems chemistry was described as "a new field of chemistry seen as the offspring of prebiotic and supramolecular chemistry on the one hand and theoretical biology and complex systems research on the other" [34]. Except for focusing on inanimate processes, systems chemistry is similar to its "uncle", systems biology [32,35–37]: it seeks insight into complex networks of interacting molecules and into how these lead to their system-level functional properties. The way in which specific interactions between components propagate through an entire system dictates emergent properties [38]. The emergence of new functions from interactions is possible because of nonlinearities and hierarchies in these interactions. Nonlinearity may not only arise from a process depending nonlinearly on the concentration of its reactants; it may also arise from interactions between two or more processes, i.e., when one process causes concentration changes that stimulate a second process, whilst that second process causes concentration changes that stimulate the former process. The former process then stimulates itself through the latter process and depends more than linearly on its own reactants. The understanding of nonlinear systems requires precise experimental data on various systems components and their interactions, and then modeling to integrate the data [37,39]. In this section, we shall discuss the chemistry of the various forms of arsenic, iron, and their interactions as well as their effects on living organisms.

*2.1. Arsenic Chemistry*

The prevalence of arsenic toxicity is not only due to trivalent inorganic arsenic [arsenite; As(III)] but also to pentavalent arsenic [arsenate; As(V)]. The concentration, form, and persistence of each of these two chemical species depend on a network of chemical reactions. At room temperature, elemental arsenic is a yellow, waxy solid that converts into grey arsenic upon exposure to light. Heated at atmospheric pressure in the absence of oxygen, elemental arsenic sublimes into a yellow gas, but heating arsenic in air will yield a white smoke instead, testifying to rapid oxidation to arsenic trioxide ($As_2O_3$), which has a garlicky odor [40,41] and binds to metal hydroxides (of ferrous iron and ferric iron, aluminium, manganese, chromium, copper, antimony, potassium, magnesium, sodium, nickel, and zinc), which are themselves subject to redox reactions and precipitation into grey matter. Grey arsenic is the usual, stable form of elemental arsenic (As). It is insoluble in water and body fluids: there is reactivity with respect to $O_2$, $H_2$, sulfur compounds, phosphate, bicarbonate, and even chloride. pH, alkalinity, *p*Ka's, redox potentials (*E*), temperature (e.g., in thermal decomposition), hydrolysis, bonding instability, the oxidation of pyrites, acid mine drainage, the grain size of the sediment, silicates, water solubilization, absorption and adsorption, levels of organic carbon, humic acid, alkanes, sulfhydryl groups, and methylation all play roles naturally in setting the activities of arsenicals. In addition to these, multiple anthropogenic factors affect arsenic toxicity: insecticides, pesticides, arsenic-based fertilizers [42], herbicides (pre-emergence and post-emergence), wood preservatives, feed additives (for poultry and swine), dyes [43], chemotherapeutic agents, silicon-based chips in micro-electronics (semiconductors), smelter-based by-products (arsenic trioxide: a glass decolorizing agent), the burning of fossil fuels, the irrigation of excess groundwater, and interactions with prokaryotes and eukaryotes. This chemical network is determined by the thermodynamic and kinetic properties of arsenic and the other chemicals mentioned, as well as by the pH, partial pressure of oxygen, and ambient redox state. The chemical interactions include oxidation–reduction and allotropic modifications. In the environment, the chemistry depends on the activities, concentrations, and growth rates of a multitude of microorganisms (see Section 3). In this review, we will first discuss the chemical network and then the biochemical one.

2.1.1. Thermodynamics of Arsenic

Ambient redox potential (*E*) (provided there is sufficient time or catalysts to enable equilibration) and pH (for which equilibration is usually very rapid) impose constraints on arsenic transformation in the natural environment [44] (Figure 2). Arsenic can occur in four oxidation states (−3, 0, +3, and +5), i.e., as $As^{3-}$ [in arsine, $AsH_3$], $As^0$ [semi-metallic arsenic], $As^{3+}$ [arsenite As(III), e.g., $H_3AsO_3$, $As_2O_3$], and $As^{5+}$ [arsenate As(V), e.g., $H_3AsO_4$] [45]. In soil, the former two states (−3 and 0) occur only under highly reducing conditions in terms of redox potential (i.e., high negative *E* in Figure 2), except for when the pH is extremely low. Under moderately reducing conditions and circumneutral pH (e.g., pH 4–10 and an ambient redox potential of around 0 V, such as may occur in anoxic subsurface waters and sediments), the trivalent and pentavalent forms of arsenic, i.e., As(III) and As(V), are thermodynamically comparable in stability (Figure 2). Under relevant conditions, the arsenite occurs as arsenous acid ($H_3AsO_3$), whilst the arsenate is either the mono- or the di- ortho-arsenate ion ($HAsO_4^{2-}$ and $H_2AsO_4^-$, respectively; Figure 2). Due to the p$Ka_1$ value around 9.5 (Figure 2), anion dihydrogen arsenite (arsonic acid, $H_2AsO_3^-$) occurs at a much more than tenfold-lower concentration [44,46]. The threshold redox potential that is required to oxidize arsenite to arsenate (indicated by the thick colored lines in Figure 2) decreases (i.e., becomes more negative) with increasing pH, meaning that for any given redox potential, arsenite is oxidized more readily to arsenate at a higher pH. Consequently, at a pH slightly lower than 7 and an ambient redox potential slightly more negative than 0 V, arsenic tends to be present as arsenite, whilst at an alkaline pH and a positive redox potential, arsenate dominates. As both arsenite and arsenate forms

are soluble in water, this should not matter for the mobility of arsenic, as long as other redox compounds such as iron are absent (but see Section 2.3).

From the classical "Pourbaix diagrams" for arsenic (Figure S2), a different, inappropriate conclusion might readily be drawn. These diagrams often compare the thermodynamics at the physical and chemical reference conditions of 1-molar aqueous concentrations [47], which favor precipitation. At these high concentrations, the precipitated form of arsenite (i.e., $As_2O_3$) is more stable than arsenous acid and the prediction would be that at pH values equal to or below 7, the arsenite should be immobile. Because aqueous arsenite concentrations are actually in the micromolar range, Figure 2 is more realistic than Figure S2, and both arsenate and arsenite are mobile. In the range where water is stable (i.e., between the two dashed red lines in Figure 2), immobile, metallic arsenic should not form either (Figure 2). In oxidizing water, i.e., at high $E$ values (e.g., $E = 400$ mV; $-E = -400$ mV in Figure 2), deprotonated forms of arsenic acid, i.e., $H_2AsO_4^-$ and $HAsO_4^{2-}$, dominate between a pH of 4 and 14, e.g., in most soil and surface water. (Upland and soils with $E < 300$ mV; i.e., $-E > -300$ mV in Figure 2 are considered anaerobic [48–50] and will rather host arsenious acid, particularly at the more acidic pHs).

An important consequence of their respective predominant electric charges is that aqueous arsenate is more readily removed from water by anion exchange, e.g., with positively charged soil material, than arsenite is. Consequently, arsenite exists in non-ionic (neutral) form ($H_3AsO_3$) in natural water (Figure 2), which renders its adsorption performance to various cationic adsorbents poor [51]. Both oxidation states of arsenic have been detected, however, under both oxic and anoxic conditions [52,53]: the thermodynamically most stable forms are not necessarily the most abundant when chemical (e.g., redox reactions with iron) or physical (e.g., rivers flowing) processes keep their redox state from equilibrating with the dissolved oxygen concentration [54].

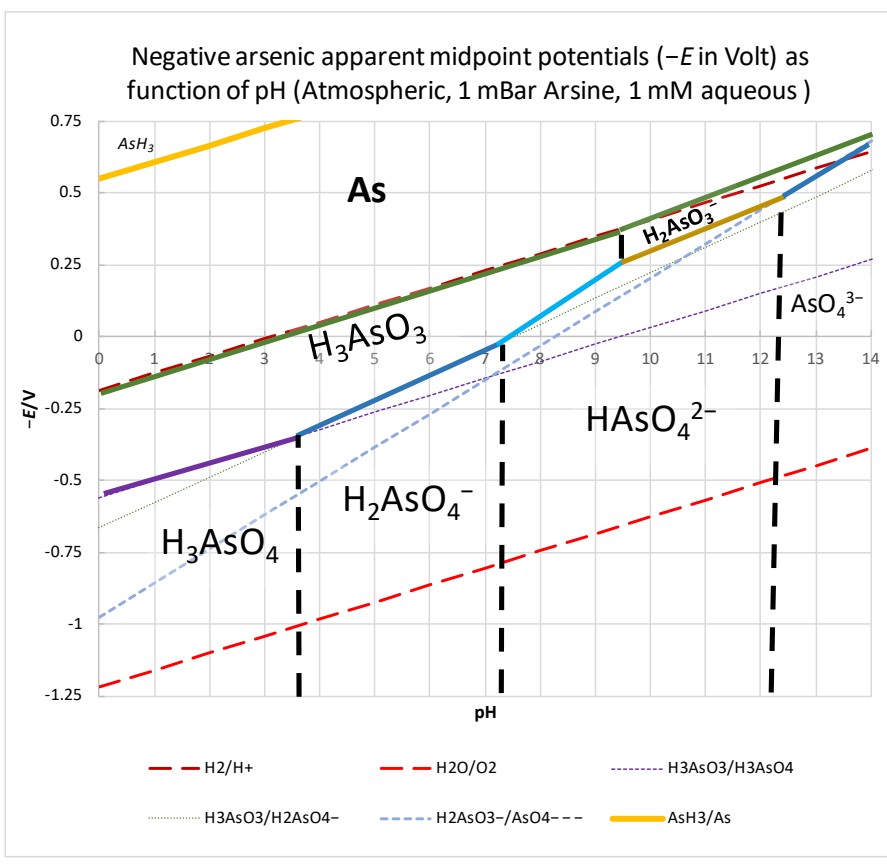

**Figure 2.** Negative redox potential versus acidity ($-E$ versus pH) "mirror Pourbaix-diagram" indicating the thermodynamically lowest Gibbs energy form of arsenic at the pH and redox potential

given by the position in the plot and this for 25 °C and 101.3 kPa and other relevant standard conditions. Rather than the physical–chemical standard conditions (Figure S2), we here used 21% $pO_2$, 0.55 µbar $H_2$, 1 mbar pArsine, and 1 mM for solutes (cf., [55]). Solid species are in bold face, gases are in italics, solutes in normal type. $-E/V$: minus the redox potential in Volt. Solid lines indicate the redox potential ($-E$) at which the bordering substances are at equilibrium with each other in their standard states at the corresponding pH. The thick, colored lines separate arsenite [As(III); on the left] from arsenate [As(V)] in their various protonation states (purple: $H_3AsO_3$ from $H_3AsO_4$; darker blue: $H_3AsO_3$ from $H_2AsO_4^-$; lighter blue: $H_3AsO_3$ from $HAsO_4^{2-}$; brown: $H_2AsO_3^-$ from $HAsO_4^{2-}$; very dark blue: $H_2AsO_3^-$ from $AsO_4^{3-}$; very dark green: As from $H_3AsO_3$; dark green: As from $H_2AsO_3^-$). Vertical dashed black lines indicate $pK_a$ values. The long-dashed red lines indicate the upper and lower edge of the relevant region for aqueous media, i.e., where water is stable versus $H_2$ and $O_2$ production, respectively. The thin dashed lines are theoretical redox potential lines partly collinear with and used to construct the other lines. As compared to the classical Pourbaix diagrams, the diagrams used here we used have been mirrored around the $E = 0$ axis, so that the ordinate shows the negative redox potential $-E$, which one may call the 'electron potential' because it indicates the chemical potential (Gibbs energy, equal to the electron potential multiplied by the Faraday constant) of the electron relative to electrons in the $H_2/H^+$ couple at pH = 0 and 1 bar hydrogen partial pressure. A high electron potential indicates that the electron wishes to jump downward in the diagram, i.e., to a redox couple of lower electron potential with the difference in electron potential corresponding to the Gibbs energy released in unit electron volt.

### 2.1.2. Mechanistic Understanding of Arsenite Toxicity

The toxicity of arsenic species varies in the following order: arsenite > arsenate > mono-methylarsonate (MMA; $CH_5AsO_3$) > dimethylarsinate (DMAA; $C_2H_6AsO_2$) [56]. Arsenite [As(III)] is 60–100 times more toxic than that in the oxidized state [57–59]. Inorganic arsenic compounds are about 100 times more toxic than DMAA and MMA.

Arsenite strongly binds to sulfhydryl groups in proteins, thereby impairing the activities of 200 enzymes [43,60]. The pyruvate dehydrogenase complex and 2-oxoglutarate dehydrogenase complex use such sulfhydryl groups in their coenzyme lipoic acid to bind and de-carboxylate their substrate (Figure 3); trivalent arsenic exerts its toxic effects mainly by inhibiting lipoic acid [61]. This reduces the influx of acetyl groups into, as well as the progress through, the TCA cycle, thereby limiting the reduction of $NAD^+$ to NADH and fumarate to succinate. NADH and succinate are two main redox substrates of the oxidative phosphorylation of ADP to ATP [62].

**Figure 3.** Scheme depicting arsenite binding to the thiol groups of lipoic acid, thereby inactivating the function of the pyruvate dehydrogenase complex. In the absence of arsenite, the binding of pyruvate by the same thiol groups leads to decarboxylation of the pyruvate.

### 2.1.3. Mechanism of Arsenate Toxicity in Living Systems

Arsenate is a structural analog of phosphate that competes with phosphate in many biochemical reactions [8,57,60,63]. The bonds that arsenate forms with phosphate and carboxyl groups are unstable and hydrolyze spontaneously: in oxidative phosphorylation, the analogue of the mitochondrial ATP synthesis reaction, i.e., ADP + arsenate + $4H^+_{out}$

→ ADP-arsenate + 4H$^+$$_{in}$, is followed by the rapid hydrolysis of the high-energy bond of the ADP-arsenate (ADP-arsenate → ADP + arsenate). Arsenate similarly uncouples substrate-level phosphorylation in glycolysis (Figure S3): 1-arseno-3-phosphoglycerate is derived from the glycolytic pathway via the bonding of arsenate and glyceraldehyde-3-phosphate, which is catalyzed by glyceraldehyde phosphate dehydrogenase (GAPDH) [64] (glyceraldehyde-3-phosphate + AsO$_4$$^{3-}$ + NAD$^+$ $\overset{GAPDH}{\rightarrow}$ NADPH + H$^+$ + 1-arseno-3-phosphoglycerate). In less than 2.5 s, 1-arseno-3-phosphoglycerate hydrolyzes spontaneously to 3-phosphoglycerate, bypassing one of the two Gibbs energy harvesting steps of glycolysis [65]. This compromises various ATP-dependent cellular processes, such as transport and signal transduction pathways [66].

Different from the effects of arsenite, the enzymes that synthesize ATP are not inactivated by the arsenate. There is indeed some consensus that arsenate is the less toxic of the two arsenics [67,68]. Arsenate may be more toxic due to conversion into arsenite rather than directly, i.e., when it engages in the uncoupling mechanisms discussed above [61]. But, as systems biology goes, it may be a combination of the two mechanisms at weights that depend on conditions.

However, again by substituting for phosphate, arsenate also alters the conformation of various proteins and small molecules and interrupts their functions, e.g., bone phosphate → bone arsenate and glucose-6-phosphate → glucose-6-arsenate. It can inactivate up to 200 enzymes, particularly phenylarsine oxide (PAO) glutathione reductase, glutathione S-transferase, glutathione peroxidase, thioredoxin reductase, thioredoxin peroxidase, DNA ligases, Arg-tRNA protein transferase, trypanothione reductase, IκB kinase β (IKKβ), pyruvate kinase galectin 1, protein tyrosine phosphatase, JNK phosphatase, Wip1 phosphatase, E3 ligases c-CBL, and SIAH1 [69].

*2.2. Iron Chemistry*

The aim of this paper is to discuss the systems chemistry of arsenic toxicity. The differential precipitation of arsenate and arsenite with ferric iron (hematite or goethite, see Section 2.3) and the variation of this with varying effective redox potential are absolutely crucial to the arsenic toxicity of groundwater. We therefore need to discuss ferrous and ferric iron, the precipitation of the latter as, e.g., hematite, and the binding of the latter to arsenate.

Figure S4 presents the thermodynamically most stable forms of iron as a function of pH and negative redox potential (i.e., the electron potential $-E$) [70]. In the $-E$ *versus pH* region where water is stable relative to water → oxygen gas and H$^+$ → hydrogen gas (the area between the dashed red lines in Figure S4), iron is most stable as the solid hematite (Fe$_2$O$_3$; its hydrated form goethite (FeOOH) also precipitates and is only slightly less stable) at pH 7 and above. Only under reductive (anaerobic) conditions (i.e., at highly positive $-E$), aqueous ferrous iron (Fe$^{2+}$(aq)) is more stable than hematite, particularly at a lower pH (e.g., $-E_{0'} = -0.2$ V at pH = 4).

The ordinate of diagrams such as Figures 2 and S4 reflects the Gibbs energy of electrons. Downward transitions dissipate Gibbs energy and can thereby occur spontaneously (i.e., without extra Gibbs energy input). At pH = 7, the negative standard midpoint potential ($-E_{0'}$) of the H$_2$O/O$_2$ couple is $-0.815$ V [71] and, under atmospheric circumstances (i.e., 21% oxygen), the actual negative redox potential is $-E' = -0.804$ V (Figure 2). In the presence of molecular oxygen and at pH = 7, metallic iron is oxidized readily to aqueous ferrous iron (Fe$^{2+}$(aq)), as two electrons then drop from the negative Fe/FeII midpoint potential of 0.55 V to $-0.815$ V in a reaction dissipating almost twice (0.55–($-0.815$))·96.5 = 130 kJ/mol electrons (a bit less if the Fe(II) concentration exceeds our reference concentration of 10 μM). On the other hand, the subsequent oxidation of the aqueous ferrous to *aqueous* ferric iron (Fe$^{3+}$(aq), as opposed to ferric iron in Fe$_2$O$_3$ or FeOOH) is *not* a process that dissipates much Gibbs energy: minus the midpoint potential of the Fe(III)/Fe(II) couple is as low as $-0.77$ V (and independent of pH), leaving only some 45 mV ($-770-(-815)$) for Gibbs energy dissipation at pH = 7. Only at more acidic pH (e.g., pH = 4), minus the apparent midpoint potential of

oxygen reduction is sufficiently far below minus the redox potential of the Fe(III)/Fe(II) ($-0.77$ V) for substantial oxidation of ferrous iron to ferric iron by molecular oxygen to occur (Figure S4; only then the lower dashed red line lies far below the blue line and its horizontal extrapolation).

But why then does iron corrode all the way to *ferric* iron oxide near neutral pH in the presence of oxygenated water? An indication of why such corrosion is still possible is the fact that at pH = 1.5, $Fe^{3+}$(aq) and hematite have the same midpoint potential with respect to reduction to $Fe^{2+}$(aq); there the blue line crosses the green line in Figure S4). Consequently half the ferric ion should there precipitates as ferrihydrite, which then reorganizes to goethite (FeO(OH)) or hematite ($Fe_2O_3$). The overall reaction for the oxidation of ferrous iron then becomes:

$$4Fe^{2+} + O_2 + 4H_2O \leftrightarrow 2Fe_2O_3 \downarrow + 8H^+ \tag{1}$$

This process is highly nonlinear in that the oxidation of ferrous iron affects pH, $pO_2$, and iron precipitation, which, in turn, affect the rate and equilibrium position of the redox process. At a less acidic pH, there is more such precipitation, as the precipitation liberates aqueous protons (Equation (1); this corresponds to the steep upward slope of the green line in Figure S4). Indeed, the solubility product of $Fe^{3+}(OH^-)_3$ is $10^{-37}$ or less [72], making the concentration of ferric iron at pH = 7 smaller than 0.1 fM (but <0.1 mM at pH = 3). Consequently, ferric iron is highly immobile at pH > 1.5 and so should any associated arsenate be, making also the latter less mobile and less available for uptake by living cells without special facilities (see Section 3.2.1.). The consequent insolubility of ferric iron compared to ferrous iron increases the *effective* negative midpoint potential appreciably, i.e., from $-0.77$ V to some +0.2 V around pH = 7 (see green line in Figure S4). Consequently, at a neutral pH and with negative redox potentials below 0.2 V (such as $-0.815$) in the presence of oxygen), aqueous ferrous iron is highly *un*stable vis-à-vis its oxidation to *precipitated* ferric iron (the green line in Figure S4 at pH7 lies much higher than the lower dashed red line). Indeed, the most stable forms of ferric iron oxide (i.e., goethite $FeO_2H$ and the slightly more stable hematite $Fe_2O_3$) [73] abound in nature.

A simplified structure of goethite is presented in Figure S5A. The dissociation of a hydroxylate ion leaves a positive charge (Figure S5B). This charge readily explains why complexes with oxyanions are formed. A side effect of this binding of anions to the hematite is that the apparent negative midpoint potential of the ferrous/ferric couple in their complexes with the oxyanions (or with Fe(III) as a precipitate) is also higher than that of aqueous ferrous/ferric iron couple ($-0.77$ V). Their citrate complex has an apparent negative midpoint potential of $-0.37$ V, for instance, and a complex of $Fe^{II}CO_3$ with $Fe(III)(OH)_3$ one of $-0.20$ V [74], which means that oxygen (at $-E = -0.81$ V) and even nitrate (at its apparent negative midpoint potential of $-0.42$ V) becomes feasible acceptors of electrons from $Fe^{2+}$ (aq). The complex with the hydroxylate ion in neutral hematite has the strongest of these effects, leading to an apparent negative midpoint potential of +0.2 V at pH 7 (Figures S4 and S6).

At pH > 7 and high ambient negative redox potential, ferrous iron tends to precipitate as magnetite ($Fe_3O_4$) instead, where one of every three Fe atoms has been reduced to ferrous iron and of which the apparent negative midpoint potential is also approximately 0.2 V at pH 7 (Figure S4). Different ratios of ferrous to ferric iron are also possible, as is rationalized by Figure S5C. Together this means that also ferrous iron can be immobilized in the iron oxyhydroxide precipitates.

That ferrous iron is more often in a soluble aqueous form ($Fe^{2+}_{aq}$) whilst ferric iron is immobile as hematite or magnetite (Figure S4) constitutes the basis for the subsurface iron removal (SIR) technology that has been applied in Europe for many decades to remove iron from groundwater [75,76]. SIR injects oxygen into the groundwater, which then oxidizes dissolved ferrous iron to immobile ferric iron.

*2.3. Interactions of Arsenic with Ferrous and Ferric Iron*

The redox state of arsenic, i.e., the arsenite/arsenate couple, and thereby arsenic mobility and toxicity through groundwater and corresponding drinking water, depends in a complex manner on a variety of processes that affect the ambient redox potential (and dissolved oxygen concentration), pH, and precipitation of arsenate and arsenite. In this review, we discuss its interaction with iron oxidation, reduction, and precipitation in some detail. This may then serve as an example of the relevance of other interactions, such as those with sulfides and sulfites. Ferric hydroxide [$Fe(OH)_3$] (ferrihydrite) and its less hydrated forms goethite ($FeO(OH)$) and hematite ($Fe_2O_3$)] play important roles in the biogeochemical cycle of arsenic as both arsenate and arsenite anions are adsorbed onto these iron oxides [77–79]. According to the electrostatic "outer sphere mechanism" illustrated for goethite by Figure S5A,E, the goethite hydrates and then dissociates hydroxide (or associates a proton without hydrating):

$$(Fe^{III}O_2H)_n(Fe^{III})_2O_4H_2 \text{ (s)} \longleftrightarrow (Fe^{III}O_2H)_n(Fe^{III})_2O_3H^+ \text{ (s)} + OH^- \tag{2}$$

Its then cationic surface (Figure S5B) binds anions like arsenate:

$$(Fe^{III}O_2H)_n(Fe^{III})_2O_3H^+ \text{ (s)} + H_2AsO_4^- \longleftrightarrow (Fe^{III}O_2H)_n(Fe^{III})_2O_3H_3AsO_4 \text{ (s)} \tag{3}$$

but much less so arsenite because the latter is electrically neutral at pH < 9. In the "inner sphere mechanism", the goethite may first undergo an energetically cheap rearrangement (Figure S5D) that brings two hydroxyl groups of the goethite into close proximity:

$$(Fe^{III}O_2H)_n(Fe^{III})_2O_4H_2 \text{ (s)} \longleftrightarrow (Fe^{III}O_2H)_n(Fe^{III})_2O_2(OH)_2 \text{ (s)} \tag{4}$$

Then, it can form a double bond with arsenic (or arsenious) acid upon a double hydrolysis reaction:

$$(Fe^{III}O_2H)_n(Fe^{III})_4O_6(OH)_2 \text{ (s)} + H_3AsO_3 \longleftrightarrow (Fe^{III}O_2H)_n(Fe^{III})_4O_6HAsO_3 \text{ (s)} + 2H_2O \tag{5}$$

Arsenite may be more prone to engaging in this "inner sphere mechanism" than arsenate because the more anionic arsenate is better solubilized by water and less prone to interact with the iron hydroxides.

The co-precipitation of arsenate with ferric iron is one of the nonlinearities that determine the mobility of arsenic in groundwater: aeration of the groundwater and the concomitant decreased ambient negative redox potential cause the oxidation of arsenite to arsenate, a process that should in itself depend linearly on the oxygen tension and dissolved aqueous oxygen concentration. However, the increased oxygen tension also causes the oxidation of ferrous iron. The ferric iron oxide precipitates, causes the co-precipitation of the arsenate, and thereby stimulates the oxidation of arsenite. The oxidation and immobilization of the mobile form of arsenic (i.e., arsenite) should thereby be nonlinearly dependent on the oxygen tension. This increases the complexity of the system.

The complexity is further increased by multiple other phenomena. One is that not only hydroxides but also protons can dissociate from the ferric hydroxide surfaces. The surface charges of ferrihydrite and goethite depend on pH, ionic strength, and the more detailed structure of the precipitate [80]. The pH of point zero charge is pH = 9.1 for goethite and 8.2 for freshly prepared ferrihydrite that had subsequently been aged for three weeks [81]. At pH > 8.5 the ferrihydrite and at pH > 9.1 goethite repel the arsenate electrostatically, leaving the "inner sphere mechanism" (chemical bonding, see Figure S5F) to effect adsorption. Accordingly, any adsorption above pH = 8.5 should be less disrupted by anions such as carbonate. That the binding of arsenate to ferrihydrite is less sensitive to high ionic strength than was expected [82] may indeed be due to the contribution of the "inner sphere" mechanism. The release of arsenic is indeed pH-dependent and related to the total iron and free iron oxides in the sediments [83]. At high pH (pH > 9, where arsenite is deprotonated), more arsenite than arsenate is adsorbed onto ferrihydrite, whilst at low

arsenic concentrations and low pH (pH = 4), more arsenate is adsorbed (see above): indeed, Dixit and Hering (2003) found more arsenate than arsenite sorption to amorphous iron oxides below pH 5–6, whereas arsenite had a stronger affinity to iron oxides above pH 7–8 [84].

A complication is the possible transfer of electrons between iron and arsenic. For a pH higher than 7, the negative iron midpoint potential ($-E_{0'}$ = 0.2 V at pH 7) is more than 0.18 V higher than that of the arsenic [As(III)/As(V)] negative midpoint potential (i.e., the electron potential $-E_{0'}$ = −0.1 V at pH 7) (Figure S6), indicating that the As(III)/As(V) ratio would be more than 100 times the Fe(II)/Fe(III) ratio if the two elements reached redox equilibrium; the arsenic would then be much more reduced than the iron (Figure S6). Consequently, incompletely oxidized iron would compromise the oxidation and immobilization of arsenic as it would reduce the arsenate. Of course, direct re-oxidation of the arsenite by molecular oxygen that may be available should remedy this situation: at ($-E_{0'}$ = −0.82 V (pH = 7), minus the $O_2/H_2O$ redox potential is still much lower than that of the arsenite/arsenate couple (−0.1 V) (horizontal green arrow in Figure S6). It is important, however, that this reaction is catalyzed and this is not assured in the absence of microorganisms (see below).

In summary, at pH ≈ 6, the anionic arsenate binds more strongly to the slightly positive ferrihydrite than neutral arsenite does and is immobilized because of the extremely low solubility products of goethite and hematite. At a highly alkaline pH, ferrihydrite may lose adsorptive power towards arsenate but still bind with arsenite. At a highly acidic pH (pH < 5), ferric iron precipitates solubilize and release arsenic.

### 2.4. Interactions of Arsenic with Soil and Groundwater Sediments

In the previous section, we discussed how arsenic may be adsorbed onto ferric (hydr)oxide precipitates. As these precipitates are components of soil, this is an important aspect of the interactions between arsenic and soil. Arsenic also interacts with calcium ions, however. Considerable precipitates form at a neutral pH, less at an acidic pH and none at an alkaline pH [85]. This process is promoted by the presence of pyrolized biomass (biochar). The impregnation of biochar with groundwater further enhances the precipitation of arsenic, which thereby boils down to the same mechanisms as discussed in the previous section. Arsenic is also absorbed by calcined (i.e., heated) magnesite (MgO) [79].

### 2.5. The Complexity of Arsenic Toxicity

The above analysis has shown that for a number of reasons, the issue of overall arsenic toxicity is complex. First, in a living organism per se, in its microbiome, or in the ecology around it, arsenate may be reduced to arsenite or methylated, whereby the toxicity changes. Second, in its actions, arsenate experiences competition from phosphate. The usual intracellular (and extracellular) phosphate concentration is much higher than that of phosphite, so arsenite should experience less such competition. Third, arsenate is more anionic than arsenite (see above), which causes the former to associate more with the (slightly) positively charged ferrihydrites (ferric oxyhydroxide $Fe(OH)_3$) or its stable dehydrated structures goethite ($\alpha$-FeOOH) and $\alpha$-hematite ($Fe_2O_3$) [82] than with its less positively charged ferrous equivalent ($Fe(OH)_2$) [73] or with the soluble $Fe^{2+}$.

Fourth, the relative sorption of arsenite and arsenate by ferrihydrite depends on pH, the former adsorbing more at an alkaline pH, possibly due to its then negative electric charge and less hydration by the water. Upon the oxidation of ferrous iron, there are then two opposite effects on the arsenic level in the water phase: (i) the drop in pH (Equation (1)) may cause the release of the then less negative arsenite and (ii) the precipitation of $Fe_2O_3$ (Equation (1)) may cause the co-precipitation of more arsenate [depending on the total number of cationic binding sites present in the $Fe_2O_3(s)$ and indirectly of more arsenite as this may also be oxidized by the molecular oxygen (Figure 2)].

Fifth, by reducing the mobility of arsenate and/or arsenite where it contacts the human population, arsenic may disappear from drinking water, so that all the above phenomena may have an indirect effect on arsenic toxicity for humans. This is the basis of the SAR process to be described below. Sixth, the actual toxicity of arsenic will differ

between organisms as well as individual humans (also between children and adults) due to differences in import and efflux systems for arsenite and arsenate. Seventh, not only pH and redox state but also the buffer capacity of aquifers and the activity of redox processes, together with amounts of ferric and ferrous irons as well as arsenite and arsenate, matter. These differ substantially between aquifers, however [86]. Butaciu and colleagues [15] performed a factor analysis of the composition of groundwater samples using a fuzzy hierarchical clustering method. They found silicate hydrolysis and carbonate dissolution to be of additional importance.

The processes mentioned above may operate simultaneously and influence each other nonlinearly. The issue of arsenic immobilization through binding to ferric iron is complex, therefore, and should be assessed experimentally under conditions close to any possible application and then analyzed profoundly, possibly with assistance from mathematical modeling and statistical analyses (see also Section 5). The issue requires systems chemistry.

*2.6. Remediation*

2.6.1. Abiotic SAR: Abiotic Arsenic Removal Strategies

In situ oxidation of arsenic and iron in aquifers has been tried in the DPHE-Danida Arsenic Mitigation Pilot Project at a few selected sites [87]. A very similar method was also introduced by Sen Gupta and co-workers in West Bengal, India, for the mitigation of subterranean groundwater arsenic [88]. Based on this process, Van Halem and colleagues (2010) introduced a cost-effective in situ technology called subsurface arsenic removal (SAR) [89] (Figure 4). It can be operated without secondary waste generation on the surface. The principle of SAR is the abiotic, in situ oxidation of iron along with arsenic by the injection of oxygenated water. No chemicals are added and microorganisms are not considered. The aquifer material merely acts as a subsurface substrate for iron precipitation and arsenic co-precipitation. Groundwater abstracted from a drinking water well is aerated in a tank and then re-introduced into the aquifer through the same tubewell by opening a valve in a pipe connecting the water tank to the tubewell pipe under the pump head (Figure 4). SAR is based on the principles discussed below:

(i).　Under circumneutral pH conditions, aqueous ferrous iron reacts abiotically with oxygen, resulting in ferric iron oxyhydroxides (ferrihydrites, Figure S4, Equations (6) and (7)):

$$4\,Fe^{II}(aq) + O_2 + 4\,H^+ \rightarrow 4\,Fe^{III}(aq) + 2\,H_2O \tag{6}$$

$$4\,Fe^{III}(aq) + 8\,H_2O \rightarrow 4\,Fe^{III}O_2H(aq) + 12H^+ \tag{7}$$

(ii).　The ferric iron oxyhydroxides precipitate onto more crystalline ferrihydrites such as hematite ($Fe_2O_3$) and goethite ($FeO(OH)$) in the aquifer's soil (Equation (8)):

$$(Fe^{III}O_2H)_n\,Fe^{III}O_3H_3(s) + 4\,Fe^{III}O_2H(aq) \longleftrightarrow (Fe^{III}O_2H)_{n+4}Fe^{III}O_3H_3(s) \tag{8}$$

(iii).　The ferrihydrite precipitate provides sorption sites for arsenate (Figure S5C,D, Equation (2) with both macromolecules hydrated by one extra water: $(Fe^{III}O_2H)_{n+4}Fe^{III}O_3H_3$ (s) $\longleftrightarrow (Fe^{III}O_2H)_{n+3}Fe^{III}O^+Fe^{III}O_3H_3$ (s) + $OH^-$)

(iv).　Arsenite can also be oxidized by the oxygen to arsenate anion at pH = 7:

$$H_3AsO_3 + \tfrac{1}{2}\,O_2 \longleftrightarrow H_2AsO_4{}^- + H^+ \tag{9}$$

(v).　and then binds to a sorption site:

$$(Fe^{III}O_2H)_{n+3}Fe^{III}O^+Fe^{III}O_3H_3\ (s) + H_2AsO_4{}^- \longleftrightarrow (Fe^{III}O_2H)_{n+3}Fe^{III}O^+H_2AsO_4{}^-Fe^{III}O_3H_3\ (s)$$

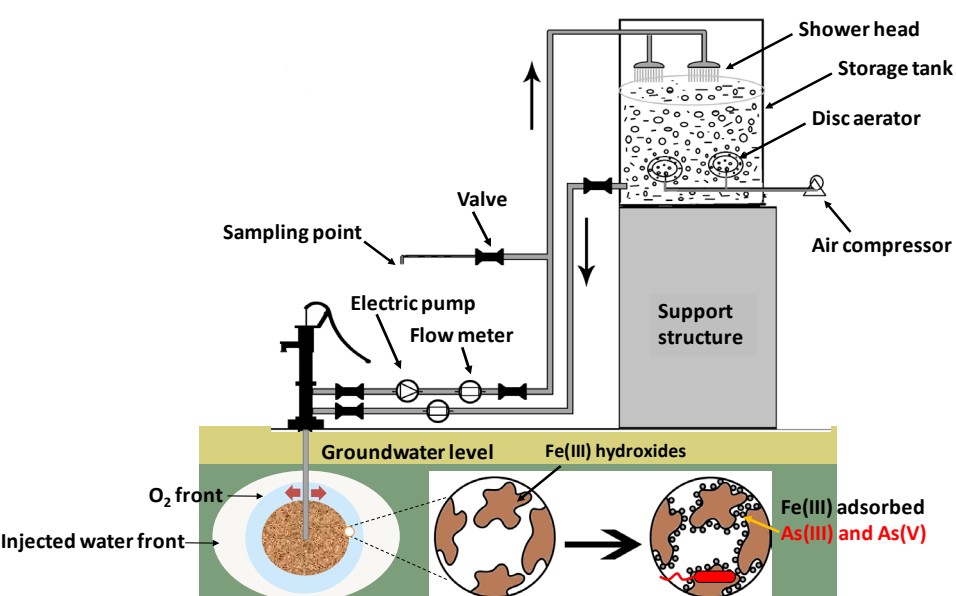

**Figure 4.** Scheme of an abiotic small-scale subsurface iron and arsenic removal (aSAR) system: abiotic oxidation of subsurface Fe(II) by the molecular oxygen in re-injected groundwater after re-oxygenation of the latter in a tank should lead to the precipitation of ferric iron oxide as geothite (FeOOH) or hematite ($Fe_2O_3$) and the adsorption thereto of ferrous iron and arsenic. The vessel on top of the support structure serves to aerate the groundwater pumped up from the aquifer. The electric pump is particularly important during the first phase of water extraction from the well—as much as 500 L. The manual pump is important for solving mechanical, technical, or electrical problems. The water tank is kept well above ground level in order to keep it out of reach of children, keep it cleaner, and assist the injection of water into the aquifer through the corresponding hydrostatic pressure. The SAR unit was operated in 55 consecutive cycles. Each cycle began at 9:00 a.m. local time with 0.5 $m^3$ volume of extraction from the SAR well, which required ~0.5 h (approximately 20 L/min). The subsequent extraction of 3 $m^3$ water volume required an additional 1.5 h. Then, the first 0.5 $m^3$ of water was aerated for 3 h and injected back into the well around 3 p.m. The remaining extracted 3.0 $m^3$ water was discarded into a nearby a lake. Thereafter, the pump was switched off until the next day. After injection, the injected water thereby stayed in the aquifer for some 18 h before the new round of extraction and reinjection on the next day.

This binding (Figure S5E,F, Equation (5)) immobilizes the arsenate, thereby keeping it away from well water that serves as drinking water [90,91].

For arsenite adsorption, the net total of all these processes is as follows:

$$(Fe^{III}O_2H)_n Fe^{III}O_3H_3 \text{ (s)} + 4\, Fe^{II}(aq) + 3/2\, O_2 + 5\, H_2O + H_3AsO_3 \longleftrightarrow$$
$$(Fe^{III}O_2H)_{n+3} Fe^{III}O^+ H_2AsO_4^- Fe^{III}O_3H_3 \text{ (s)} + 8\, H^+ \tag{10}$$

Any remaining aqueous ferrous iron not reacting with oxygen tends to reduce arsenate to yield more magnetite, hematite or goethite, and arsenite (the black arrow lies higher than the green arrow in Figure S6). The magnetite may be re-oxidized by oxygen to become hematite and the arsenite may be oxidized to arsenate, which may adsorb onto the hematite (Equation (10)). SAR technology may be extended by setting the levels of other abiotic factors, such as pH, $SiO_4$, $HPO_4^{2-}$, and $HCO_3^-$, which may affect the success of SAR [89,92,93].

### 2.6.2. Nitrate and SAR

At pH 7, the arsenite/arsenate redox couple has an apparent negative midpoint potential of $-E_{0'} = -60$ mV (Figure 2, green arrow in Figure S6), i.e., lower than that of ferrous iron/hematite ($-E_{0'} = 0.2$ V; black arrow), but higher than that of cytochrome c

($-E_{0'} = -0.21$ V) and certainly high enough to reduce nitrate ($-E_{0'} = -0.41$ V; blue arrow in Figure S6) or molecular oxygen ($-E_{0'} = -0.81$ V; red arrow) [55,71]. Indeed, nitrate ($NO_3^-$) injected into arsenite-contaminated groundwater lowered the aqueous arsenite concentration significantly [94]. Presumably, the immobilization of arsenic is enhanced through arsenite oxidation coupled to the biological reduction of nitrate, the resulting arsenate being more absorbed onto iron oxides than the arsenite (see above and [94]), and possibly being reduced back to arsenite by ferrous iron with the generation of more goethite and hematite. This illustrates the potential of nitrate for immobilizing arsenic in anoxic environments: its effects may be both direct (through arsenite oxidation) and indirect (through ferrous iron oxidation and then co-precipitation with arsenic). Again, we see that processes interact nonlinearly, with a possibly enhanced dependence of arsenic mobility on the ambient redox potential.

### 2.6.3. Abiotic SAR Is Not Yet Effective

In practice, abiotic SAR performance has been disappointing. Van Halem et al. (2010) [89] found that the removal of arsenic was not as tightly coupled to iron oxidation as suggested (but not really proven) by Equations (6)–(10). In a study by Freitas et al. [92], the arsenic level of groundwater was reduced by SAR, but could not be brought below the WHO guideline of 10 µg/L. Bicarbonate and phosphate appeared to compromise SAR. Rahman et al. [95] found arsenic adsorption to be limited kinetically, suggesting that the oxic phase should be prolonged or catalyzed. Even though nitrates in the aquifer (e.g., due to the use of fertilizers nearby) may inhibit the reduction of ferric iron and thereby increase the robustness of its arsenate absorption, in the presence of excess reducing equivalents such as those in methane, ferric iron may still be reduced, leading to the release of arsenic [96]. The systems chemistry complexity increases further when it is acknowledged that the oxygenation of subsurface groundwater may release arsenic from arsenic sulfide in the soil [97,98].

### 3. The Water Chemistry of As and Fe Connects to Microbiology: Biology Does Matter

At atmospheric oxygen pressure, the oxidation of aqueous ferrous iron to ferric iron and the precipitation of the latter are processes that occur at substantial rates in the absence of living matter. We know this as rust formation. At consequently reduced dissolved oxygen concentrations in groundwater, the rate of these abiotic processes is reduced proportionally however, unless microorganisms are present that catalyze them. Some microorganisms can harvest Gibb energy from the oxidation of ferrous iron. Provided that other elements required for their growth are present, these microorganisms can amplify to high abundance so that the iron oxidation at reduced oxygen levels again becomes substantial until the oxygen activity becomes really low. Other microorganisms can similarly oxidize arsenite, reduce arsenate, reduce ferric iron, or alter ambient redox potential (oxygen concentration) and pH in processes that are all very slow in the absence of microorganisms. Thus, arsenic and iron chemistry in natural waters is not determined just by iron and arsenic chemistry itself but also to a considerable extent by the microbial activities that occur and develop in those waters.

Conversely, the abundance of these microorganisms depends on the availability of the various forms of iron and arsenic, first as sources of Gibbs free energy (e.g., through the oxidation of ferrous iron or arsenic by molecular oxygen catalyzed by these organisms) for their growth and second in terms of the toxicity of arsenite and arsenate, which causes death or growth inhibition. Precipitated ferric oxides and associated arsenic species may further enable microorganisms to attach and profit from a stable source of these materials or from the possibility to deposit toxic compounds outside their cells onto such material.

The integration of these processes in actual aquifers has the effect that the physical–chemical processes around arsenic and iron influence the concentrations of arsenate, arsenite, ferrous iron, and ferric oxides, as well as the pH and ambient redox potential. The changes in these concentrations, pH, and redox potential affect the metabolism, growth, and

death of microorganisms that themselves affect the levels of iron, arsenics, pH, and ambient redox potential and thereby again abiotic iron and arsenite oxidation. Consequently, the physicochemical processes around iron and arsenic in actual aquifers must be discussed in the context of the active microbiology in these aquifers, and, conversely, the microbiology must be discussed in the context of the physical chemistry of arsenic and iron. In the present section (Section 3), we shall discuss the microbiological contribution to arsenic and iron chemistry. In Section 4, we shall then discuss the integration of the physicochemical and microbiological processes for an actual case of dealing with the arsenic toxicity of groundwater and drinking water.

### 3.1. Organic Matter

The co-occurrence of very high concentrations of dissolved organic matter (DOM) with elevated concentrations of dissolved arsenic and iron in reductive groundwater has often been observed [99]. DOM significantly influences arsenic biogeochemistry, and *reactive* organic matter facilitates the microbial release of arsenic from sediment to groundwater [57]. Sedimentary organic matter may provide carbon sources for microorganisms. Moreover, shallow groundwater is usually recharged by surface water, importing reactive organic carbon and accelerating microbial processes. Autotrophic microbial growth may further increase the organic matter density. Such changes in the organic matter potentially influence both the spatial and the temporal evolution of groundwater arsenic geochemistry.

### 3.2. Microbiology of Arsenic: What Can Microorganisms Do?

Microorganisms cannot perform miracles. What they do must be consistent with thermodynamics, i.e., the processes that they catalyze must run downhill in terms of Gibbs energy. They can, however, escape from this limitation by coupling a thermodynamically uphill reaction to a different, thermodynamically downhill reaction. The most important example is the coupling of the often thermodynamically uphill reaction of microbial growth [100] to a process delivering Gibbs energy. The latter process is photon absorption in photoautotrophs, the catabolism of organic material in heterotrophs, and inorganic reactions such as arsenite or ferrous oxidation by oxygen or nitrate in lithoauthotrophs. This example is most important because microbial growth leads to the autoamplification of the chemical activities. The catalyst of the process, i.e., the microorganism, can thereby become tremendously active. Limitations here are the time it takes for the microbes to replicate and the other chemicals they require for growth. The elements carbon, nitrogen, phosphorous, and sulfur are minimally required, and this is an important issue as some ecosystems are lacking in one or more of these and, in other ecosystems, competing microorganisms strongly reduce their levels. It is an important observation that the elements and Gibbs energies may be limiting, but not the catalytic activities; the microbial geosphere is rich and dispersed enough to catalyze virtually anything that is possible in terms of thermodynamics and element conservation. And it will augment itself through proliferation.

Accordingly, microbial communities drive the global biogeochemical cycling of arsenic, and they do this through diverse metabolic functions [31,101,102]. In addition to Gibbs energy harvesting and coupling, microorganisms have evolved a variety of mechanisms to overcome the effect of metal(loid) toxicity. These include (i) mechanisms that restrict arsenic entry into the cell or enable active extrusion, (ii) enzymatic detoxification through redox transformations, and (iii) chelation or precipitation [103–105]. Frequently metal(loid) resistance genes are located on mobile genetic elements and are readily transferred between different bacteria via horizontal gene transfer [106]. Taxonomically diverse bacterial populations *viz. Alpha-*, *Beta-*, *Gamma-proteobacteria*, *Firmicutes* (*Bacillus* and relatives), *Actinobacteria*, Bacteroidetes, etc., play roles in the bio-geochemistry of arsenic-rich groundwater [107–114]. Chemolithotrophic and heterotrophic arsenic-transforming bacteria deploy an array of metabolic routes in arsenic speciation, distribution, and cycling in aquatic systems [101,115–117]. Diverse microbial genes encode metabolic processes involved in arsenic-oxidation, -reduction, and -methylation [118–121] and thereby affect

arsenic speciation, mobilization, and availability as well as ecotoxicity [122,123]. Levels of mobile arsenic in groundwater depend on the balance between all the biochemical processes mentioned. These therefore need to be evaluated in any particular case of arsenic contamination in groundwater.

### 3.2.1. Microbiological Processes Benefitting Arsenic Remediation: Metal Oxidation

In principle, microorganisms could contribute to arsenic removal, in particular, ferrous iron oxidizers and arsenite oxidizers. Relevant aspects of the biochemistry, physiology, and ecology of these microorganisms are discussed in this section.

Microbial Oxidation of Ferrous Iron

Ferrous iron [Fe(II)] is an electron donor to a wide range of iron-oxidizing bacteria, and such iron oxidation can be operated at both acidic and neutral extracellular pHs, under either oxic or anoxic conditions (Figure 5). Fortin et al. noted that microbial iron oxidation is accelerated through a variety of mechanisms [124]. At pH = 4, the apparent negative midpoint potential of the oxygen–water couple amounts to $-E_{0'} = -1.0$ V, i.e., 0.23 V lower than that of the ferrous/aqueous ferric iron couple ($-E_{0'} = -0.77$; Figure S6). This implies that respiration with molecular oxygen as an electron acceptor, ferrous iron as an electron donor, and aqueous (i.e., soluble) ferric iron as a product is thermodynamically feasible in environments with an acidic pH, i.e., for *acidophilic* organisms; precipitation into goethite or hematite is not required for these energetics. Thus, ferrous iron constitutes a good source of Gibbs free energy for aerobic *acidophilic* prokaryotes [125,126]. *Acidithiobacillus* spp., (*β-Proteobacteria*) are by far the most studied group of bacteria capable of gaining Gibbs energy from the oxidation of ferrous to ferric iron at a very low pH. Edwards and coworkers (2000) isolated an iron-oxidizing archaea closely related to *Thermoplasmales* from an extremely acidic environment (pH 0.5) [125].

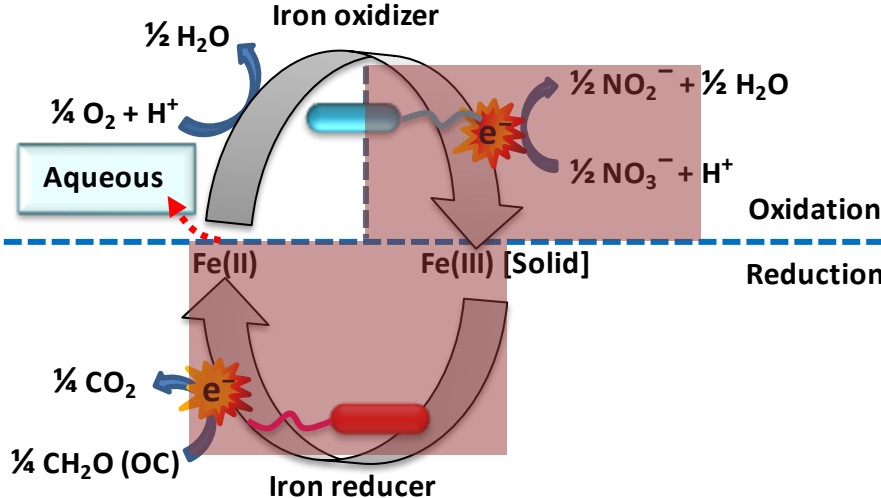

**Figure 5.** Microbial redox cycling of iron. In microaerophilic conditions, chemolithoautotrophic bacteria use Fe(II) as a source of electrons and couple the reduction of nitrate (or oxygen) to the oxidation of Fe(II) to Fe(III), which then precipitates (Fe(III)[solid] is equivalent to ferrihydrites such as hematite and goethite). In the bottom half of the cycle, heterotrophic Fe(III)-reducing bacteria couple the reduction of Fe(III) to the oxidation of organic carbon (OC), whereupon Fe(II) is released to the water as $Fe^{2+}$(aq). The overall reaction is the oxidation of organic carbon by nitrate or oxygen to carbon dioxide and water, nitrite or nitric oxides, and even nitrogen, which yields Gibbs free energy in the form of ATP, which drives the biosynthesis and replication of the microbes. The shaded areas indicate the absence of oxygen.

This explains the first of the four physiological groups of bacteria that oxidize ferrous iron, i.e., (i) acidophilic, aerobic iron oxidizers; (ii) anaerobic, photosynthetic iron oxidiz-

ers [74]; (iii) neutrophilic, microaerophilic iron oxidizers; and (iv) neutrophilic, anaerobic (nitrate-dependent) iron oxidizers.

Fully anaerobic ferrous iron oxidation is conducted by anoxygenic, phototrophic, purple, non-sulfur bacteria utilizing ferrous iron as a reductant for thermodynamically uphill carbon dioxide fixation, with light as a Gibbs energy source [127]. It is clear where the photosynthetic iron oxidizers get their Gibbs energy from, but how can we understand the energetics of the two remaining groups?

The aerobic oxidation of ferrous iron by neutrophilic microorganisms (Figure 5) may seem paradoxical as at pH = 7 the midpoint potentials of the ferrous/ferric iron couple (the extended blue line in Figure S4) and the oxygen–water couple are too close to allow for Gibbs energy to emerge in the process. However, the precipitation of the ferric iron as ferrihydrite reduces the effective midpoint potential to ($-E_{0'}$ = 0.2 V at pH = 7 (Figure S4), much lower than the electron potential $-E_{0'}$ = 0.8 V of the oxygen–water couple. Microaerophilic conditions are required because only then can aerobic, ferrous iron-oxidizing bacteria compete effectively with the abiotic oxidation of iron by oxygen that would dominate at atmospheric oxygen tensions [128]. Neutrophilic, microaerophilic conditions are common where iron-rich waters meet an oxic-anoxic interface due to low mixing rates and the limited molecular diffusion of oxygen in water [129]. Microaerophilic, iron-oxidizing bacteria have been found to thrive in wetland soils, plant rhizospheres [130,131], places where iron seeps into groundwater supplying freshwater [132,133], and drinking water distribution systems [134]. Chemolithoautotrophic bacteria (e.g., *Gallionella* spp. and *Sideroxydans* spp.) extract their metabolic energy from iron oxidation under these conditions [135–137], but this is not the case for obligate heterotrophs such as the *Sphaerotilus-Leptothrix* groups [138]. Among the four recognized species of *Leptothrix*, *L. ochracea* is the only species for which there is circumstantial evidence for autotrophic growth using Gibbs energy derived from iron oxidation [74]. Aerobic, chemolithotrophic, magnetite-oxidizing bacteria may contribute significantly to ferrous iron oxidation at a circumneutral pH [129,132,133,136,139,140] (Figure S4). Currently, all known oxygen-dependent, neutrophilic, chemolithotrophic iron oxidizers belong to the *Proteobacteria* group, with *Gallionella* as the best known representative [137], belonging to the *β-proteobacteria* group. *Gallionella* sp. can also grow on organic compounds such as glucose, fructose, and sucrose [141], and sulfur ($-E_{0'}$ = 0.06 V) or sulfide ($-E_{0'}$ = 0.10 V) [142,143] may serve as better electron donors compared to the ferrous iron in cases where ferrihydrite precipitation is slow or problematic.

A signature of iron-oxidizing bacteria is the unique morphological structures they produce, such as sheaths (in heterotrophic species) and helical, stalk-like filaments (in autotrophic species, although autotrophic *Siderooxydans* spp. form neither sheaths nor stalks [74]). Excreted from the cell surface, the stalk of *Gallionella* acts as an organic matrix for the deposition of the ferrihydrites produced (e.g., as hematite $Fe_2O_3$) [144]. In view of the thermodynamic importance of Fe(II) oxidation to ferrihydrite (see above), these unique structures are not just morphological but essential for the energetics. Moreover, arsenate may be trapped by Fe(III), which binds to the stalk or other extracellular polymeric substances (EPS) on the surface of bacteria to form As(III)-Fe(III)-EPS complexes [145,146], or just by Fe(III) in magnetite (see above and Figure S5).

Twisted stalks of *Gallionella ferruginea* may further act as a protective mechanism against precipitated ferric iron or oxygen toxicity [147]. The metals may also bind as cations to the cell surface in a passive process [124], perhaps again with tighter binding of the triply charged ferric iron, thereby again increasing the iron's electron (negative redox) potential $-E_{0'}$. Many neutrophilic, iron-oxidizing bacteria can form ferric iron minerals that can co-precipitate with arsenic [148]. Also, heterotrophic *Leptothrix* strains are able to deposit iron oxyhydroxides onto their cell surface [149].

The biological oxidation of ferrous iron in the absence of oxygen and in dark subsurface waters is also possible by light-independent chemoautotrophic microbial activity using nitrate as the electron acceptor [150] (Figures 5 and S6). Indeed, nitrate-reducing, iron-oxidizing bacteria are the most important catalysts for the generation of ferric oxides under

anaerobic conditions [128]. Nitrate-dependent, iron-oxidizing microorganisms are able to oxidize both soluble and insoluble ferrous iron minerals [151]. For thermodynamic reasons, the ferric iron must occur in complexes (such as with hydroxide in ferrihydrite/magnetite): nitrate/nitrite $-E_{0'} = -0.41$ V at pH = 7 and $-E_{0'} = -0.60$ at pH = 4 are both too low for reduction by free ferrous iron transiting to free ferric iron ($-E_{0'} = -0.77$) (Figure S6). Even *Escherichia coli* is capable of nitrate-dependent iron oxidation [152].

In a further demonstration of how interactions between various inanimate and animate processes may accomplish processes that are otherwise thermodynamically impossible, there are at least three further solutions to the small Gibbs energy yield of iron respiration with nitrate as electron acceptor. One is the use of complexed ferrous iron as a substrate: *Thiobacillus denitrificans* oxidizes ferrous sulfide (FeS; negative midpoint potential at pH 7 of approximately ($-E_{0'} = -0.25$ V), i.e., higher than the $-E_{0'} = -0.42$ V of the nitrate/nitrite couple; Figure S6 [150]. This nitrate-dependent iron sulfide oxidation has since been demonstrated for the hyperthermophilic archaeon *Ferroglobus placidus* [153], the mesophilic *Proteobacteria Chromobacterium violacens* [154], and the *Paracoccus ferrooxidans* strain BDN-1 [155]. A further alternative is the nitrate reduction ($-E_{0'} = -0.42$ V) or chlorate reduction ($-E_{0'} = -0.79$ V) [156] coupled to ferrous iron oxidation in the presence of carbon and a Gibbs energy source, which has been documented for the heterotrophic *Dechlorosoma suillum* strain PS [157] as well as for the *Acidovorax* strain BoFeN1 [158].

There have been several reports describing nitrate-dependent ferrous iron oxidation ($-E_{0'\text{FeII/Fe2O3}} = 0.2$ V; we assume oxidation to hematite or goethite, which stabilizes the ferric iron tremendously and thereby stimulates ferrous iron oxidation, see above) by *Geobacter metallireducens* with a reduction of nitrate to nitrite ($-E_{0'} = -0.42$ V; Gibbs energy yield of 60 kJ/mol electrons). The generation of nitrite enables the further reduction to NO ($-E_{0'\text{NO2}^-/\text{NO}} = -0.37$ V; $\Delta_r G' = -55 \frac{\text{kJ}}{\text{mol\_electrons}}$) and then to N$_2$O ($-E_{0'\text{NO/N2O}} = -1.17$ V; $-132 \frac{\text{kJ}}{\text{mol\_electrons}}$) [74,154,159], i.e., addressing negative midpoint potentials ($-E'$s) much lower than that (0.2 V) of the donor couple Fe(II)/Fe(III) in ferrihydrite (but higher than the $-E_{0'} = -0.77$ of the Fe(II)/soluble Fe(III) couple), i.e., that the ferric iron resides in the ferrihydrite rather than being dissolved in water makes an enormous difference. The reduction may even continue with that of nitrous oxide to produce the very stable molecular nitrogen ($-E_{0'\text{N2O/N2}} = -1.3$ V) [160] in the complete denitrification pathway [161]. That this *delta-proteobacterium* can use the Gibbs energies of all these reactions to support its growth has not been ascertained, but its abundance in anaerobic sediments [162] might suggest this. *Acidovorax ebreus* controls nitrate-dependent, anaerobic iron oxidation through nitrite formation from nitrate and the subsequent abiotic reduction of nitrite by additional ferrous iron. Anaerobic iron oxidation may be widespread in the environment [151].

A remaining thermodynamic issue is how organisms respiring ferrous iron with molecular oxygen (or nitrate) are able to engage in the standard biochemistry found in almost all living organisms [163]. At pH 7, the negative midpoint potential of NAD(H) is $-E_{0'} = 0.32$ V and that of FAD(H$_2$) is $-E_{0'} = 0.22$ V [55], which are both more positive than that of the Fe(II)/hematite couple ($-E_{0'\text{FeII/Fe2O3}} = 0.2$ V; Figure S6), so that they cannot be reduced by the latter. This engagement may be helped by so-called "electron bifurcation": the Gibbs energy of electrons flowing from ferrous iron to molecular oxygen is then used to drive proton pumps and is thereby partly stored as a protonmotive force ($\Delta p$). The latter then drives reverse electron transport from ferrous iron to the intracellular redox coenzymes NAD and FAD [164].

Microbial Oxidation of Arsenite

Oxygen as an electron acceptor

The biological oxidation of arsenite (Figure 6) has been recognized as an attractive alternative to its abiotic oxidation due to its potential specificity for arsenite, efficiency, effectiveness at lower oxygen tensions (through a low $K_M$), and cost effectiveness in addition to environmental friendliness [165]. In environments where significant amounts of arsenite

are oxidized to arsenate within a short period of time, this oxidation can be attributed to arsenite-oxidizing bacteria [166].

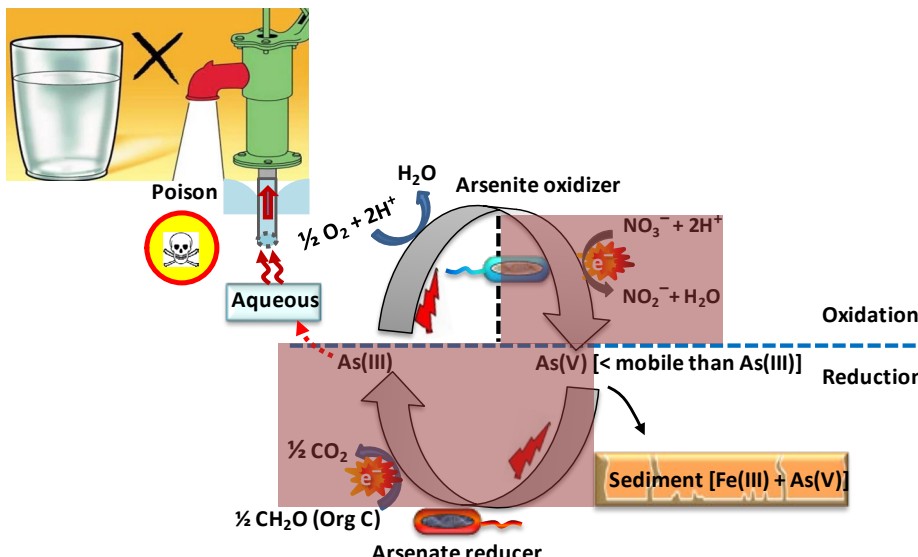

**Figure 6.** Scheme of putative microbial arsenic cycling in aquifers and consequences for drinking water. Microbial arsenite oxidation (above the dashed line) is mediated by a number of chemolithoautotrophs under aerobic conditions at the surface or anaerobic conditions below that surface, using oxygen or nitrate, respectively, as the terminal electron acceptor. Microbial arsenate reduction (below the dashed line) is mediated by dissimilatory, arsenate-respiring bacteria coupling arsenate reduction under anaerobic conditions to the oxidation of organic carbon, the resulting arsenite entering into aqueous solution. The shaded areas indicate the absence of oxygen.

Despite or perhaps precisely because of its biochemical toxicity (see above), arsenite is readily converted by a diversity of prokaryotes. Arsenite-oxidizing bacteria are classified into heterotrophic (HAO) and chemolithoautotrophic (CAO) arsenic oxidizers [120,167]. Heterotrophic arsenite oxidation may serve primarily as a detoxification reaction, rather than as a Gibbs energy source (in heterotrophs, other catabolic reactions readily provide this Gibbs energy): it converts arsenite encountered in the cell's periplasmic space into the less toxic arsenate, perhaps making it less likely for the arsenic to enter the cell [101]. CAOs, on the other hand, couple the oxidation of arsenite to the reduction of oxygen (Figure 2) with the aim of capturing some of the 0.8 V (Figure 2) redox potential difference as Gibbs energy for carbon dioxide fixation and cellular growth [168]. CAOs have also been reported to grow heterotrophically however [168].

More than 50 phylogenetically diverse, arsenite-oxidizing (auto- and heterotrophic) species, distributed over 25 genera, have been isolated from various environments, especially mesophilic ecosystems [169]. Green, for instance, reported arsenite-oxidizing bacteria stemming from cattle dipping baths [170] and Battaglia-Brunet et al. isolated a *Leptothrix* sp. strain S1.1 from the settling pond sediments of mine drainage that was able to oxidize 0.1 g/L of As(III) in 1 week at 12 °C [171]. Phylogenetically, arsenite oxidizers are dispersed within the *Alpha-*, *Beta-*, and *Gamma-proteobacteria*; *Actinobacteria*; *Firmicutes*; and *Deinococcus-Thermus*. Green sulfur bacteria (e.g., *Chlorobium limnicola* and *Chlorobium phaeobacteroides*) and filamentous green non-sulfur bacteria (e.g., *Chloroflexus aurantiacus*) may also be capable of arsenite oxidation, as homologs of the gene-encoding arsenite oxidase (see below) have been identified in their genomes [60,172–175]. The most extensively studied heterotrophic arsenite oxidizer is *Alcaligenes fecalis* [120]. Little is known regarding the role of archaea in the oxidation of arsenite.

Heterotrophic *Alcaligens faecalis* [176] and *Pseudomonas pudia* [177] have not been shown to extract Gibbs energy from the oxidation of arsenite during heterotrophic growth.

There is one known exception: *Hydrogenophaga* sp. str. NT-14, a β-*proteobacterium*, can oxidize arsenite whilst it grows heterotrophically, its arsenite oxidation still being coupled to the reduction of oxygen and yielding extra Gibbs energy for growth [121]. Gihring and Banfield (2001) isolated a peculiar thermophilic species of *Thermus* (strain HR 13) from an arsenic-rich hot spring. Under aerobic conditions, it was able to oxidize arsenite apparently for detoxification purposes, i.e., without conserving Gibbs energy. However, under anaerobic conditions, strain HR 13 can grow on lactate using arsenate as its electron acceptor, reducing it to arsenite [178].

Arsenite oxidase, located on the outer surface of the inner bacterial membrane, has been identified in both autotrophic and heterotrophic bacteria [101,120]. The enzyme is the first component of an electron transport chain that enables arsenite to reduce oxygen to water in a process coupled to proton pumping and the subsequent generation of ATP from ADP and phosphate (see Supplementary Materials, Figure S7). The genes encoding arsenite oxidase (*aio* genes) show considerable divergence; the *aioA* sequences of CAOs are phylogenetically distinct from those of HAOs [169,179]. Only two putative arsenite oxidase genes have been identified in *Aeropyrum pernix* and *Sulfolobus tokodaii* by sequence homology searches of their published genomes [180].

### Alternative electron acceptors

Molecular oxygen is poorly soluble in water (up to some 0.25 mM only, and also the rate at which it dissolves is small whenever the surface-to-volume ratio is small). Aerobic microbes in the upper oxic layers of aquifers consume dissolved oxygen, maintaining anaerobic zones below them. Anaerobic or facultative anaerobic microbes thereby become dominant in the underlying anoxic environment [181]. Alternative oxidants (e.g., nitrate; $-E_{0'} = -0.42$ V) then have the potential to support growth through the microbial oxidation of arsenite $-E_{0'} = -0.06$ V (Figures 2 and S6). Several studies have indeed demonstrated that anaerobic microorganisms can engage in nitrate-dependent arsenite oxidation to gain Gibbs energy [167,182]. Such arsenite-oxidizing, denitrifying bacteria have been isolated from various environments and enriched [53,167,183,184]. Besides nitrate, chlorate ($ClO_3^-$; $-E_{0'} = -0.79$ V) [185] can be an oxidant (electron acceptor) for the anaerobic microbial oxidation of arsenite. *Dechloromonas* sp. strain ECC1-pb1 and *Azospira* sp. strain ECC1-pb2 constitute examples [182].

Most arsenite-oxidizing, denitrifying organisms are *Alpha*, *Beta*, or *Gamma-proteobacteria*. The first identified anoxic, arsenite-oxidizing bacterium was *Alkalilimnicola ehrlichii* strain MLHE-1, a haloalkaliphilic facultative chemolithoautotroph: it is also able to grow heterotrophically with acetate ($-E_{0'} = 0.29$ V for $CO_2$/acetate couple; [71]) as its electron donor, either aerobically, or anaerobically with nitrate as an electron acceptor. A novel type of arsenite oxidase gene (*arxA*) was identified in the genome of this extremophile, which fills a phylogenetic gap between the arsenate reductase (*arrA*) and arsenite oxidase (*aioA*) clades of arsenic-metabolizing enzymes [186]. Anoxic, chemolithoautotrophic, arsenite-oxidizing strains DAO1 and DAO10 (closely related to *Sinorhizobium* and *Azoarcus* sp., respectively) living under "normal" environmental conditions are also able to oxidize arsenite to arsenate with complete denitrification of nitrate (see above for the energetics) [53].

### 3.2.2. Microbiological Contribution to Arsenic Mobilization
Metal Reduction

Microorganisms can play a role in toxic arsenic release indirectly via the oxidation of organic carbon coupled to the reduction of arsenic-bearing iron oxyhydroxides. This then causes the opposite of the SAR process, i.e., dissolution of the arsenic-bearing iron oxyhydroxides and the subsequent release of arsenic in the more mobile arsenite form [187–189]. Microorganisms may also cause arsenite release directly via the utilization of arsenate as an electron acceptor [77,190]. An important factor in both processes is the organic matter that is used as an electron donor for metal reduction by the indigenous microbial community in aquifers. These heterotrophic activities may impact SAR negatively.

Microbial Reduction of Ferric Iron

Microbial iron reduction is one of the most significant mechanisms for the oxidation of natural organic matter or organic contaminants to carbon dioxide in diverse aquatic environments [162]. It alters the geochemistry of submerged soils and sediments, as well as that of surface and subsurface water [191]. The microbial reduction of ferric oxides can have the following significant effects on water quality and soil chemistry [192]: (a) an increase in water-soluble iron concentration ($Fe^{2+}$ being more soluble than $Fe^{3+}$ in ferrihydrites); (b) a pH decrease (Equation (1)); (c) cation displacement from exchange sites that become less negatively charged; (d) increased solubility of phosphorus, arsenic, and silica because their complexation partner $Fe^{3+}$ disappears [193]; and (e) the formation of new minerals such as magnetite from hematite (Figure S4). The increase of dissolved ferrous iron in groundwater affects the taste of drinking water and causes staining, which can be expensive to remediate [194]. As we have seen in Section 2.3, it may also affect the whereabouts and toxicity of arsenic, in another example of the complex interactions of different processes.

Members of the iron-reducing family *Geobacteraceae* dominate aquifers where ferric iron reduction is a significant terminal electron-accepting process, especially in the presence of organic matter as an electron source (at pH 7 and negative redox potentials around $-E = 0.4$ V, whilst the $Fe^{2+}$/goethite $-E_{0'}$ is 0.2) [190,195,196]. In these environments, these members dominate the degradation of organic matter and control the mobility of toxic metals [197]. Yet, iron reducers are phylogenetically and physiologically diverse. They are distributed widely among bacteria (mostly belonging to *Proteobacteria*, *Firmicutes*, *Actinobacter*, *Bacteroidetes*, *Fusobacteria*) and archaea [198,199]. Most of the iron-reducing archaea are hyperthermophilic, some are mesophilic or thermophilic methanogens [200].

*Thermatoga marinetime* and *Pyrobaculum islandicum* conserve Gibbs energy from hydrogen oxidation ($-E_0$ at pH 7 = 0.4 V, red arrow in Figure S6) by ferrous iron precipitating as hematite ($-E_{0'}$ at pH 7 = 0.2 V; black arrow in Figure S6) [201]. For this the hydrogen partial pressure needs to be higher than our standard of 0.55 µbar in Figure 2, or the pH should be smaller than 7. At pH > 6, the Gibbs energy gain is small however unless the hydrogen partial pressure exceeds 1 mbar. Based on the thermodynamic possibilities, Fe(III)-reducing microorganisms can be divided into two groups. Fermentative bacteria use ferric iron as an electron sink only, which can help to generate their ATP via substrate-level phosphorylation during acetate production [202]. The other group is often more important in environmental iron reduction [203] and iron cycling [162,204] in aquatic sediments, submerged soils, and subsurface anoxic environments. Its members gain Gibbs energy via oxidative phosphorylation. This is driven by electron transfer from organic matter ($-E_{0,glucose/CO2'} = 0.43$ V; purple arrow in Figure S6) through NADH at electron potential $-E_{0,NADH/NAD}' = 0.32$ V, through an electron transport chain to aqueous Fe(III). The process is called dissimilatory iron reduction. This multi-enzyme process accounts for the valence transition of iron from the ferric $[Fe(III)_{sol}]$ to the ferrous $[Fe(II)]$ form, which it cannot only couple to this oxidation of organic matter but also to that of hydrogen (effective electron potential at pH 7 of $-E_{0'} = 0.25$ V, Figure 2, but higher ($-E' = 0.4$ V; red arrow in Figure S6) at higher partial pressures of hydrogen gas). These are indeed thermodynamically downhill processes that could energize microbial growth. In practice, however, the ferric iron exists in, or as aqueous ferric iron in equilibrium with, ferrihydrites such as goethite and hematite. Then, its negative midpoint potential at pH = 7 is as high as $-E' = 0.2$ V rather than $-E' = -0.77$ V (Figure S4), leaving only little Gibbs energy for dissipation or harvesting. In other words, the free aqueous Fe(II) concentration in equilibrium with ferrihydrite precipitates is extremely low so that its reduction is thermodynamically difficult and slow. Direct access to the ferric iron in the ferric oxyhydroxide precipitates should therefore be important to speed up the process.

The precise mechanisms of microbe-mediated, dissimilatory iron reduction have remained elusive. An important issue is indeed this lack of solubility and mobility of ferric iron, which tends to precipitate with whatever oxyanions are available. Iron-reducing organisms may cope with the difficulty of transferring electrons from the cell to insoluble

iron minerals by at least three mechanisms (Figure 7) [203]: (i) by having physical contact with iron minerals via the formation of conductive cell surface appendages called pili or nanowires between the cell and the surface of the minerals and a functioning Fe(III)-reductase located in the outer membrane; (ii) by using electron shuttling compounds produced endogenously or acquired exogenously; and (iii) by producing ligands or using extracellular chelators that aid in the dissolution of the solid-phase ferric oxide, generating dissolved ferric iron that should be more available to the microorganism [90,91]. Yet, none of these three proposed solutions solve the thermodynamics problem: these catalytic mechanisms cannot increase the Gibbs energy difference between the organic matter and the Fe(II)/hematite couple. Perhaps the recently proposed gear shifting mechanism offers a solution to this predicament [205], with a transfer of two electrons coupled to the pumping of a single proton or with an increase of the $H^{+\rightarrow}$/ATP stoichiometry of the proton translocating ATPase.

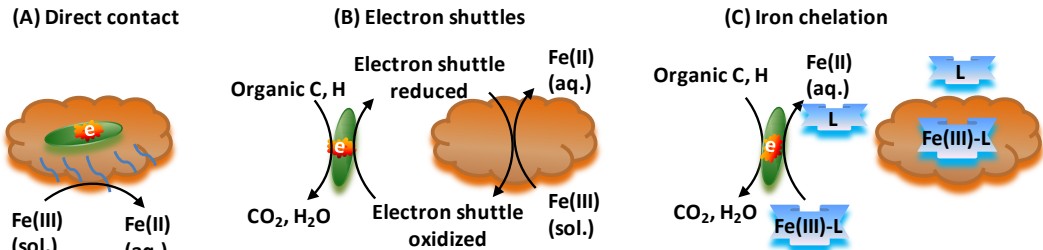

**Figure 7.** Mechanisms of iron reduction and possible interactions between microorganisms (green oval shapes) and iron oxides (cloudy brown shapes). (**A–C**) illustrate the mechanisms of iron reduction by means of direct contact by cells, extracellular electron shuttles, and chelation by the iron-ligand L, respectively. (**A**) In *Geobacter* spp., direct contact with the oxide surface is required. Nanowires, conductive extracellular appendages, facilitate electron transfer by functioning as an electrical conduit to the Fe(III) oxide surface. (**B**) An endogenously or exogenously produced electron shuttle mediates electron transfer to solid-phase Fe(III) oxides. (**C**) The production of complexing ligands, as in the case of *Geothrix* sp., aids in the dissolution of the solid-phase Fe(III) oxide, providing a soluble Fe(III) form more readily available to the microorganism. e, electrons; L, ligand (adapted and modified from [90,91]).

Microbial Reduction of Arsenate

Two different arsenate reductases are encoded by *ars* and *arr* genes, which are linked to cellular detoxification and respiration mechanisms, respectively [123]. We shall here discuss these mechanisms separately.

Microbial Reduction of Arsenate for Arsenic Detoxification

One group of microorganisms reduces arsenate as part of a mechanism for arsenic detoxification and resistance. They do not gain Gibbs energy from this process but invest it [101,183,206]. Total flux here should be commensurate to arsenate leakage (or entry though a phosphate transporter) into the cell, rather than to the electron transfer flux required to energize growth. Therefore, arsenic detoxifiers contribute relatively little to arsenate reduction compared to dissimilatory, arsenate-respiring microorganisms.

The upper part of Figure 8 depicts a model for this *ars*-dependent arsenate resistance. Since arsenic does not play any metabolic or nutrimental role, microorganisms lack specific *arsenic* uptake systems [207]. As arsenate has structural similarity to phosphate, it enters the cell through phosphate uptake channels (Pst or Pit). Similarly, as As(III) has a structural similarity to glycerol, it can enter cells through the glycerol transport system, mainly facilitated by the aquaglyceroporin channel GlpF encoded by the *glpF* gene [208]. Once in the cytoplasm, arsenate first binds to the anion site in the ArsC, leading to the formation of an arsenate thioester intermediate. This intermediate is reduced in two phases by glutaredoxin and glutathione, leading to the formation of an intermediate Cystic2-S-As(III).

This intermediate results in the release of arsenite upon hydrolysis [63,209]. The arsenite is released from the cell via the ArsAB pump [206,210] or sequestered in intracellular compartments, either in conjugation with glutathione or other thiols or as free arsenite [211]. An arsenite chaperone (ArsD) and an ATPase (ArsA) interact with ArsB to provide high levels of arsenite resistance through the hydrolysis of ATP, presumably by powering the efflux pump further. An aquaglycerol porin gene (*aqpS*), normally associated with arsenite import, was found in place of *arsB* in the *ars* operon of *Sinorhizobium meliloti* and functioned in arsenite export [212]. AqpS channel facilitates the function of an arsenite efflux pump that is used as a substitute for the transporter ArsB. AqpS also has the ability to sensitize the cell to the arsenite in the external environment, after which the ArsC protein will reduce arsenate in the internal environment [123]. The ArsC proteins can be divided into two families of bacteria: (i) the $ArsC_{ec}$ family, which uses glutaredoxin as an electron source [213,214], and (ii) the $ArsC_{sa}$ family, which uses thioredoxin as an electron source [215,216] and requires the presence of thioredoxin reductase and NADPH to complete the catalytic cycle [217]. The ars operons can also be coupled with other ars-related genes to allow the detoxification of organo-arsenicals [208].

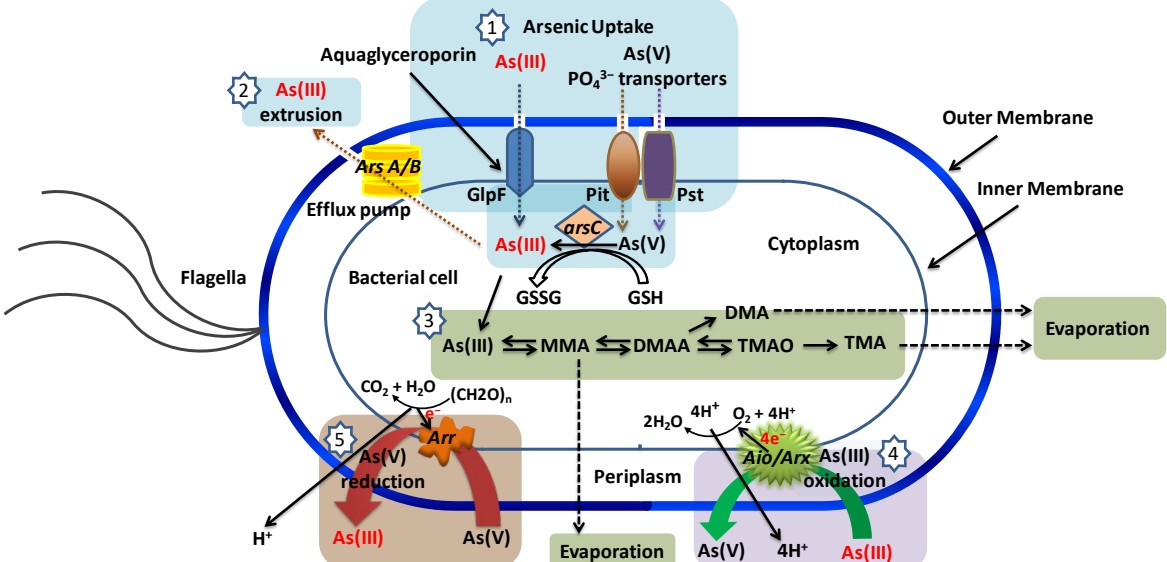

**Figure 8.** Scheme of microbial arsenic metabolism: rectangular boxes shaded with light blue, olive green, light purple, and light tan color indicate inorganic arsenic detoxification, organic arsenic methylation and detoxification pathway, autotrophic/respiratory arsenite oxidation, and respiratory arsenate reduction pathways, respectively. The bottom part of the cell shows two alternative Gibbs energy extraction processes. (1) Arsenic (As) enters the cell through the phosphate transporters [arsenate, As(V)] or the aquaglyceroporins GlpF [arsenite, As(III)]; (2) once inside the cells, As(V) is reduced by an arsenate reductase, ArsC, to As(III), which is extruded from the cell by the specific membrane pump Ars(A)B (brown dotted arrow)—this arsenic detoxification process is not coupled to proton pumping and consequent ATP synthesis; (3) inorganic arsenic can be transformed into organic species via a cascade of methylations (small black arrows); (4) arsenite enters the periplasm (dotted black arrow) via GlpF (aquaglyceroporin) and serves as an electron donor to AioA and thereby to oxygen via the As(III) oxidase AioAB or ArxAB, which produces protonmotive force as the electrons are channeled to electron acceptors such as oxygen or nitrate (broad curved green arrow), transfer negative charge across the membrane, and are coupled to proton pumping (long black arrow); see also Figure S7; (5) arsenate enters the periplasm via GlpF (aquaglyceroporin) and the extracellular/periplasmic As(V) is then used as an electron acceptor during respiration of arsenate rather than oxygen (broad curved red arrow) by the dissimilatory arsenate reductase ArrAB, which also produces protonmotive force. The processes indicated do not necessarily all occur in every single microorganism.

The reduction of arsenate to arsenite in the course of detoxification may seem counterproductive: the less toxic As(V) is converted to the more toxic As(III) before efflux; perhaps the As(III) efflux system evolved first under reducing environments and was subsequently coupled with As(V) reduction to accommodate As(V) toxicity once the earth's atmosphere became more oxidized [59,123]. The development of oxygenic conditions may then have driven the evolution of arsenate reductases due to increased arsenate levels [218]. Another interpretation relates to the effect that the various processes have together through their interactions: in the actual ecosystem, ferric hydroxides such as goethite may be present, to which arsenate is bound, which then equilibrates with toxic intracellular arsenate. The reduction of this arsenate to arsenite diminishes the arsenic binding to the ferric oxide precipitates and thereby mobilizes the arsenic, allowing for its diffusion away from the endangered organism.

Microbial Reduction of Arsenate in Arsenic Respiration

As predicted by its apparent midpoint potential around 0 mV at pH = 7 (Figure 2), arsenate can be used as a terminal electron acceptor in heterotrophic anaerobic respiration, thereby releasing Gibbs energy that can be used to support growth (Figure S6). Microorganisms performing this process are referred to as dissimilatory, arsenate-reducing prokaryotes [101]. These microorganisms can have a significant impact on the mobilization of adsorbed forms of arsenate via the conversion of the latter to the more toxic and less adsorbing arsenite [219,220]. Some arsenate reducers are also capable of iron reduction, which dissolves the co-precipitation of arsenic with iron oxides. Ferrous iron may, in turn, reduce arsenate, even abiotically (Figure S6). Other arsenate-reducing microorganisms are not capable of iron reduction [102].

The respiratory arsenate reducers are phylogenetically diverse, ranging from mesophiles to extremophiles living at extremes in terms of temperature, pH, or salinity. The bacteria include Gram-positive strains (*Desulfosporosinus* sp. Strain Y5) and *Epsilon*, *Delta*, and *Gamma* groups of *Proteobacteria* and archaea, suggesting that arsenate reduction is a widespread and evolutionarily old process [52]. These microorganisms often support their growth through the reduction of a variety of other electron acceptors including selenite (electron potential $-E_{0'} = -0.53$ V), iron ($Fe^{3+} \rightarrow Fe^{2+}$: $-E_{0'} = -0.77$ V, but see above), nitrate (/nitrite: $-E_{0'} = /0.42$ V), nitrite (/NO: $-E_{0'} = -0.37$ V), manganese ($MnO_2 \rightarrow Mn^{2+}$; $-E_{0'} = -0.40$ V), and oxygen ($-E_{0'} = -0.81$ V) [221,222]. Many of the known arsenate-respiring bacteria are heterotrophic and capable of using simple organic acids such as lactate, pyruvate, formate, fumarate, succinate, malate, and acetate as a carbon source and electron donor [223]. A few chemoautotrophic, arsenate-respiring prokaryotes can also use hydrogen as an electron donor and carbon dioxide as a carbon source [224].

The *arr* operon encodes the respiratory reduction of arsenate (Figure 8, bottom left). Arsenate respiratory reduction is mediated by a periplasmic molybdenum containing subunit ArrA, which receives electrons from ArrB (an iron–sulfur-containing subunit), which itself accepts electrons from heterotrophic catabolism through an electron transport chain including Cmya [225]. *Cym*A encodes a *c*-type tetraheme cytochrome [226] that is also required for the functioning of other terminal electron acceptors such as oxides of Fe(III) and Mn(IV)] [227].

## 4. Systems Microbiology

As described above, arsenic levels in groundwater depend on a multitude of microbiological and geochemical factors. Dependence on any of these factors depends on the prevalence of many of the others. The specific growth rate of some important species depends on local pH and arsenate concentrations, which depend on the activity of other organisms that reduce ferrihydrite and thereby solubilize arsenate. This is typically a case where networks of biochemical processes with nonlinear interactions determine functional outcomes: in such cases, systems biology may be of help. Systems biology [32,35,37] examines the emergent properties of microorganisms that arise from the interplay of genes, proteins, other macromolecules, small molecules, organelles, and the environment [228].

Microorganisms are ideal candidates for systems biology research because they are often relatively easy to manipulate and play critical roles in health, environment, agriculture, and Gibbs energy provision. One of the potential applications of systems microbiology is the management of pollution control and bioremediation; water and soil quality management systems may thereby be optimized. The following fundamental questions could then be addressed [228]: Which species are present? What are they doing? Where are they doing it? What is the environmental impact of the community? What happens to the community in the event of a natural or society-generated disturbance? And, finally, how could we reduce arsenite levels?

Systems microbiology may also identify existing or design novel microbes that can be used to address environmental, agricultural, or medical problems [39]. By modeling the metabolic and regulatory networks of common laboratory inhabitants like *Escherichia coli*, synthetic systems biologists can now build novel gene circuits that respond to new signals in a predictable way. The resulting "designer microbes" have a number of potential applications, including the degradation of persistent toxic chemicals that would otherwise poison soils and water supplies [39,228]. Engineered bacterial strains have also been used as microbial factories for generating ethanol as biofuel, feed additives, and pharmaceuticals [229–231]. The microbial production of these materials can be more cost-effective than production by traditional methods [232–234].

In the subsequent sections, we shall identify the questions that the systems microbiology approach to arsenic pollution needs to address. In order to become concrete, we shall focus on the drinking water in the country that is most troubled by the arsenic contamination of groundwater (see above), but all aspects that we shall discuss are important generally.

*4.1. Drinking Water Wells and SAR: A Case Study*

4.1.1. Which Microbes Are Present in Drinking Water Wells?

A cultivation-independent survey of 24 arsenite-contaminated drinking water wells reported large differences in microbial communities within and between groundwater samples [109]. The major bacterial community members comprised *Hydrogenophaga*, *Acidovorax*, *Dechloromonas*, *Acinetobacter*, *Aminobacter*, *Sinorhizobium*, *Pseudomonas*, *Geobacter*, *Sideroxydans*, *Gallionella*, methanogens, methylotrophs, and sulphate reducers. Sequences most closely related to heterotrophic, iron-oxidizing *Leptothrix* sp.; anaerobic, denitrifying, iron-oxidizing bacteria; and the iron-reducing genera *Albidiferax*, *Desulfuromonas*, and *Shewanella* turned up. Bioinformatics analyses suggested that iron- and arsenic-oxidizing bacteria coexist in nearly all the investigated aquifers. Iron- and arsenate-reducing microorganisms also appeared to be present in these aquifers. This rich potential may allow iron and arsenic cycling (between their two redox forms).

In cultivation-dependent analyses of the same samples under conditions requiring iron oxidation or iron reduction activities for persistence, iron oxidizers and iron reducers were again found [235]. Whilst a *Gallionellaceae*-specific PCR revealed only a limited persistence of *Gallionella*, which is a well-known iron oxidizer, a significant number of *Comamonadaceae*-related 16S rRNA gene sequences were detected. According to these criteria of persistence through extensive serial cultivation, several *Comamonadaceae* (e.g., *Hydrogenophaga* and *Acidovorax* sp. and *Rhodocyclaceae* (*Dechloromonas* sp.) appeared to engage in iron oxidation [235]. Several strains of *Hydrogenophaga*, *Acidovorax*, and *Dechloromonas* spp. are indeed known to be capable of both iron [151,159,236] and arsenite oxidation [121,183,237]. In keeping with this, *aioA* sequences were identified that most closely related (>94% amino acid identity) to those identified on the basis of the cultivation-independent analysis [235] of the same water samples.

The arsenite-oxidizing enrichments [108] recovered the additional AioA phylotypes *Paracoccus* sp. SY, *Bosea* sp. WAO, *Hydrogenophaga* sp. Cl3/*Thiobacillus* sp. S1/*Ancylobacter* sp. OL1, and *Achromobacter* sp NT-10/*Alcaligenes* sp. S46. Arsenite oxidase (AioA) containing *Hydrogenophaga* and *Acidovorax* dominated the 24 arsenic-contaminated drinking water

wells. The arsenite oxidizers that were identified included facultative anaerobes as well as facultative chemolithoautotrophs. This suggests that they can grow and oxidize arsenite under both aerobic and anaerobic conditions—in the latter conditions, probably coupling to nitrate reduction (see above). Heterotrophic bacteria oxidize arsenic but use organic carbon as a Gibbs energy source. This could be important for bioremediation purposes. For the more organic carbon is consumed by the heterotrophic iron and arsenite oxidizers, the lower the probability of dissemination of iron and arsenite into the environment through the activity of iron- and arsenate-reducing microbes powered by that same organic carbon.

Next, the microbiome diversity was investigated in water and sediment samples of an experimental SAR well, again through 16S rDNA amplicon sequencing analysis [238]. Almost 300 candidate microbial species (we shall here use the word "species" for operational taxonomic unit, OTU) were identified and attributed to 16 different phyla [238]. The dominant phylum *proteobacteria* came in the five classes *Alpha-*, *Beta-*, *Delta-*, *Gamma-*, and *Epsilon-proteobacteria*, where *Betaproteobacteria* were the most abundant in terms of number of OTUs. Genes for arsenite oxidation, i.e., *aioA* and *arrA* for arsenate reduction or close homologs thereof, resided in the aquifers, according to PCR and sequencing. We conclude that groundwater from many locations may contain genes and organisms that may well affect SAR.

### 4.1.2. Does SAR Affect the Microbial Community?

In the best of all SAR scenarios, the microbial communities would adapt positively to SAR in the sense of amplified levels of ferrous iron and arsenite oxidizers. The increased ferric iron would precipitate as hematite, goethite, or some other ferrihydrite and absorb the arsentae, thereby removing more and more arsenic from the well water.

There is evidence that such adaptation took place: *Epsilonproteobacteria* were completely absent from the SAR well, whereas the *Gammaproteobacteria* were quite abundant there. In the control wells, arsenate-reducing *Epsilonproteobacteria* (*Sulfurospirillum* sp.) were almost absent but other members of *Epsilonproteobacteria*, e.g., sulfur-oxidizing species, were present in higher abundance. Accordingly, there were *significant* ($p < 0.05$) differences between the potential metabolic types of microbial communities in reference versus SAR wells. Yet, the statistical significance was limited, possibly due to the enormous variation within each well type. In addition, physicochemical parameters changed during the treatment in the SAR well. Also, the subsurface water flow may have affected the microbial communities, causing differences between the reference and SAR wells during the injection and abstraction (average extraction speed of 0.02 $m^3$/min) of water as the injection and extraction points were in close proximity to each other (~2.5–3 m distance).

### Aerobic Iron and Arsenic Oxidizers

Contrary to the optimal scenario, however, *Gallionellaceae*-related iron-oxidizing (FeOx) bacteria, observed frequently in the reference well, were much less abundant in the SAR well (6%) and the tank (5%) water than in the reference well. *Gallionella*-related organisms were not identified in comparable SAR experiments, neither in Bangladesh nor in Mexico [89,239]. Most probably, the injected aerated water was toxic to the mostly microaerophilic, iron-oxidizing bacteria. The observation that iron was reduced in the water pumped out of the SAR wells suggests that, in addition, most of the bioavailable form of aqueous ferrous iron might have been oxidized to Fe(III) and then precipitated quickly and abiotically as solid ferric iron, leaving no substrate for the development of iron-oxidizing microbes. Therefore, the standard SAR experiment shifted between two extremes: a brief, fully oxic environment during injection of the 0.5 $m^3$ of aerated water and then a fully anoxic environment subsequently before and during extraction of the subsequent 3.5 $m^3$ water volume, most of which was derived from the anaerobic environment around the well.

Diverse microbial communities with high functional redundancy are generally more resistant to changes in oxygen levels [240]. Such functional redundancy in *Geobacteraceae*, especially *Geobacter sulfurreducens*, might have allowed for a quick response when envi-

ronmental conditions, such as exposure to oxygen, changed. Tolerance to oxygen varies among *Geobacter* species [241]. Yet, iron-reducing *Geobacter* were not observed at high abundance in the SAR well [109], although they were present throughout the experiment at low quantities. This implies that the substantial amplification of the positive scenario was not happening, notwithstanding the expected functional redundancy.

Elevated concentrations of nitrate were found during SAR in the aerated tank water but not in the same (SAR) water before aeration. This nitrate may derive from ammonia oxidation in the tank by ammonia-oxidizing bacteria due to the many paddy fields around the study well where farmers use urea as fertilizer. Indeed, illumina 16S metagenomic sequencing revealed ammonia-oxidizing microorganisms, i.e., *Nitrosomonas* spp. Ammonia monooxygenase (*amo*A) genes were identified in both the tank water and the reference well water (no cycles were being operated in the reference well, which was simply a control well without any treatment). After reinjection into the well, the nitrate may have been used as an electron acceptor by anaerobic iron oxidizers after the oxygen had run out, explaining its disappearance from the SAR water pumped up from the well. Indeed, nitrate-reducing, iron-oxidizing sequences emerged from the SAR wells [238]. In situ, these might oxidize iron into ferrihydrite precipitates. This would be consistent with the rapid abiotic reduction of the injected oxygen by ferrous iron, followed by an anaerobic phase with the reduction of nitrate by ferrous iron.

Ferric Iron and Arsenite Reducers

Under anoxic conditions, the reduction of ferric iron or arsenate is a potential Gibbs energy source too, provided it can be coupled to the oxidation of organic carbon (Figure S6). After exposure to the oxygen in injected aerated groundwater, there should be excess oxidized iron and some arsenate available for such reductions and, hence, for the amplification of resident iron reducers. Importantly, and different from the iron oxidation phase, this reductive phase is unlikely to proceed abiotically at any substantial rate: it would require the amplification of iron reducers. Both iron- and arsenate-*reducing* microorganisms were indeed found in much larger quantities in the SAR well (post-SAR) sediment than in the reference well sediment [238], suggesting that the iron in the well sediment was indeed sufficiently available for reduction by microorganisms to drive their amplification. We here witness another nonlinear system of processes, i.e., ferrous iron-oxidizing organisms, with abiotic iron oxidation reducing the oxygen tension potentially down to zero and ferric iron-reducing microorganisms then taking over and mobilizing ferric iron as ferrous iron, with consequences for arsenate release as arsenite, again depending on the levels and specific growth rates of arsenic-metabolizing microorganisms. One might have expected to find organisms specialized in robustness with respect to oxygen levels. *Shewanella* sp. can reduce iron and enhance arsenic mobility both under aerobic and anaerobic conditions [203,242,243], but they were not identified in the SAR well [235].

The presence of the iron reducers may not only have influenced the effectiveness of the SAR process negatively but it might also pose a risk should SAR be discontinued. Then, these organisms might revert back to the SAR process, removing precipitated ferric iron oxides associated with arsenic and thereby causing the re-emergence of the latter in the well water at even higher levels than before the first SAR.

Contrary to the absence of evidence of biotic ferrous iron oxidation, there is evidence of the biotic oxidation [238] of *arsenite* under SAR conditions, and the key genus is anaerobic, arsenite-oxidizing *Dechloromonas*. A major portion of the microorganisms *Acinetobacter*, *Sphingomonas*, *Flavobacterium Pseudomonas*, *Methylomonas*, and *Deinococcus* in the SAR and tank water of the experiments by [235,238] was potentially arsenic-resistant, more so than the microorganisms in the reference well. Some of these genera, i.e., *Acinetobacter*, *Methylomonas*, and *Pseudomonas*, were detected in the subsequent chemolithoautotrophic, arsenite-oxidizing enrichments [108]. Arsenic-resistant organisms *Acinetobacter* and *comamonas* sp. can extrude arsenic upon aerobic arsenate reduction, but few of them also convert arsenite to arsenate as a mode of detoxification [244,245]. Banerjee et al. reported in 2011

that *Acinetobacter lwoffii* strain RJB-2 exhibited siderophore production ability [246]. Arsenic could be mobilized from its co-precipitate with hematite or goethite due to the extraction of iron from the latter by the siderophores. Alternatively, the siderophores could support the emergence of extracellular goethite, with the concomitant precipitation of arsenic as a resistance mechanism. Therefore, the ability of strain RJB-2 to produce siderophores also provided additional explanations for developing the arsenic resistance mechanism in bacteria [247]. Importantly, however, arsenic-accumulating and -transforming bacteria should be available for bioremediation [246], with options for promoting the ferrihydrite–arsenate co-precipitation scenario.

Many OTUs in the water sampled from the SAR wells and tank water were related to methanotrophs and *Aquabacterium* (which is a facultative aerobe able to use oxygen or nitrate as electron acceptors, associated with denitrifying Fe(II)-oxidizing sediments) [245,248]. Using microcosm experiments and hydrogeochemical and microbial community analyses, Glodowska et al. demonstrated that methane functions as an electron donor for methanotrophs, triggering the reductive dissolution of arsenic-bearing Fe(III) minerals and mobilizing arsenic into the water [249]. Some of the methanogenic archaea that can accelerate arsenic release in groundwater aquifers into the methanogenic zone are highly resistant to arsenic [250]. The biomethylation of arsenic transforms inorganic arsenic to organic arsenics that evaporate (Figure 8) [251]. In the SAR wells more than in the reference wells, 16S rRNA sequences were found that were homologous to those of the archaea *Methanosarcinales* [238], which have been found to be associated with arsenic in groundwater in China [250].

### 4.2. An Assessment: Why Was SAR Ineffective?

Considering the abiotic and biotic principles and data together, we would suggest that the sediment but not the water of the SAR wells contained an appreciable amount of arsenic- and iron-cycling bacteria. To the extent that the bio-remobilization enrichment experiment performed by [235,238] mimicked the SAR well itself, the results of the SAR experiments suggest that the aerobic phase is dominated by abiotic iron oxidation by the injected oxygen. This may be followed by a biotic phase of further oxidation of ferrous iron by denitrifying organisms that use nitrate derived from agricultural urea (fertilizer) through ammonia oxidation by microbes in the water tank. The ferric iron will have precipitated as goethite or similar ferrihydrites. In parallel, there may have been arsenite oxidation to arsenate, which then adsorbed onto the ferrihydrites. After all, SAR does remove some arsenic from the water; the issue is that it does not reduce the arsenic concentration sufficiently (Freitas et al. [92]). When the oxygen and nitrate run out and the well thereby achieves reducing conditions, iron- and arsenic-reducing microorganisms may benefit and re-reduce the iron and arsenate, causing the dissolution of some of the ferrihydrite precipitates and the adsorbed arsenate. These processes are not completed in the standard SAR cycle; the organisms are too low in abundance and do not appear to amplify much. While during SAR operation itself, the reductive phase might be a minor factor reducing SAR efficiency, it constitutes a potential threat for the dissemination of iron along with arsenic into subsurface environments around the well, which should be anaerobic. Indeed, arsenic-contaminated aquifers abound in iron-reducing *Geobacteraceae*, suggesting that such a reducing condition exists in all or most of them [109,235] and that this should therefore be a concern. All in all, SAR efficiency appears to be the net effect of a great many nonlinearly interacting processes, such as high initial oxygen tensions during the injection of oxygenated water into the aquifer, which are toxic for some iron- and arsenite-oxidizing microorganisms; the subsequent abiotic oxidation of ferrous iron removing the oxygen for oxygen-tolerant ferrous iron and arsenite-oxidizing organisms that might otherwise thrive, thereby varying the abundance of these types of microorganisms; the limited and spurious influx of alterative electron acceptors such as nitrate; the re-reduction of arsenate and/or ferric iron once the oxygen is depleted; and the perhaps slow release of arsenite from the co-precipitate with ferrihydrites.

The oxidation of ferrous iron by the injected oxygen appears to be the only abiotic process in the SAR cycle; all other processes need to be catalyzed by microorganisms and it appears that all these biotic processes are slow in comparison with abiotic iron oxidation. Accordingly, one should perhaps increase the biology and make use of the fact that one could in principle achieve this specifically by stimulating or injecting organisms of choice. One should stimulate the arsenite oxidizers, particularly strains that affix to ferrihydrite substrates and deliver the arsenate there for adsorption. Here, sustained oxygenation— but at low rates so as to keep the oxygen tensions low—or the provision of nitrate as an alternative electron acceptor for arsenite oxidizers might help. Once the oxygen runs out or this oxygenation is halted, the water should be pumped out quickly from the aquifer in order to allow little time for the iron- and arsenate-reducing organisms to react. This is an example of a strategy that would pay attention to the many nonlinear effects that may occur in aquifers with growing microorganisms.

At present, SAR involves the extraction of 3.5 m$^3$ of well water, of which only the first 0.5 m$^3$ is used to fill the aeration tank, whilst the other 3 m$^3$ is discarded to a nearby lake. The 0.5 m$^3$ in the tank is aerated and subsequently reinjected into the aquifer. As a consequence, every cycle, 3 m$^3$ of groundwater is sucked into the aquifer below the well from the environment, with potentially lots of arsenic stemming from that environment. It might pay off, therefore, to oxygenate the groundwater around the drinking water well or increase the size of the tank to a capacity of 3.5 m$^3$ and fill it with additional groundwater pumped from sites away from the drinking water well, as was achieved by [95]. It might be worth storing the first 0.5 m$^3$ of extracted water in a separate tank where the concentration of arsenic and iron coincide with the WHO guideline and can be used for drinking and other purposes. The next 0.5 m$^3$ water could be stored in a separate tank for aeration for subsequent injection into the subsurface.

### 4.3. bSAR: Systems Microbiology Contributing to SAR

Above, we described abiotic SAR and discussed how it was not as effective as desired. As shown in the extensive discussions of microbial capabilities, microorganisms could in principle assist in many SAR processes. Some microorganisms are able to oxidize ferrous iron to ferric iron. Others oxidize arsenite to arsenate. Yet others take up arsenate, reduce it, and extrude arsenite. And yet others might help Fe(III) oxide precipitation by the formation of biofilms or increase the level of Fe(II) around the well. Microorganisms also have the ability to amplify their own activity by growing once the conditions are favorable. Functioning SAR might thereby emerge over time. Once a biotic SAR process could operate a little, it could amplify itself and also be robust against many types of perturbation. In addition, pre-grown microorganisms could be added to an already existing SAR process in a process called bioaugmentation. Or, a microbial community could be added to an essentially abiotic SAR process to initiate a biotic SAR.

In examples of the *a*SAR of drinking water wells, such a highly active *b*SAR has not yet emerged by itself (e.g., [107]). Apparently, the conditions were not yet optimal; a further systems microbiology analysis of the situation may be needed. Here, we shall make a start.

Confining ourselves to a biotic version of existing SAR technology, a biotic SAR process would have microorganisms in the aeration phase that would help oxidize the Fe(II). It would also have both Fe(II)-oxidizing and As(III)-oxidizing organisms below the well. Around the well, it might have Fe(III) reducers.

Arsenite-oxidizing microorganisms are widely distributed in arsenic-contaminated aquifers in South and South East Asia [109,245,252–255], China [174], West Bengal [119,256], and Taiwan [104,257], and active when provided with the proper conditions in the laboratory [107]. Arsenite oxidase gene (*aioA*) sequences most closely related to those of arsenite- and iron-oxidizing *Acidovorax* sp. abounded in the arsenite-oxidizing enrichments, but other organisms found may also have catalyzed these processes. The data indicate diverse metabolic potential for the bioremediation of arsenite in groundwater, consisting of



bioconversion to arsenate, which should then co-precipitate with ferric iron if the latter is present, e.g., due to SAR.

Microorganisms can transform arsenite to less toxic and less mobile arsenate forms; hence, the microbial oxidation of arsenite has a major impact on the natural attenuation of arsenic pollution by decreasing its bioavailability and removing arsenic from mobile soil or water environments, provided that iron also occurs in the ferric state as ferrihydrite. Also, biotic SAR depends on the oxidation of Fe(II) in groundwater by cycles of extraction, aeration, and subsequent re-injection of extracted groundwater. Arsenite and, even more so, arsenate should then co-precipitate with precipitating Fe(III) oxides. The latter are immobilized by binding to immobile elements of the soil. The arsenate appears more liable to co-precipitate with newly formed Fe(III)oxides than with already existing Fe(III)oxide precipitates. Therefore, a continuous operation of SAR may be required that continues to draw more Fe(II) from the environment of the well. Also, the pH should be monitored as at the more acidic pH caused by the oxidation of ferrous iron, arsenate will precipitate with the ferrihydrite, whereas at the more alkaline pH that may arise when oxygenation stops, more arsenite may precipitate onto the ferrihydrite (see above). The presence of calcium ions and biochar or actual organic soil material may further complicate the outcome (see Section 2.4).

In principle, microorganisms could assist in many of these processes, if only to make them more robust. Some of the organisms identified in [107–109,235] could be of interest for in situ or ex situ bioremediation methods for arsenic. The detection of arsenite-oxidizing bacterial *aioA* sequences in arsenic-contaminated Bengal delta plain (BDP) aquifers in India are indicative of their presence in this type of environment [256], and the distributions of bacterial communities based on *aioA* and 16S rRNA sequences are congruent to our studies. Cavalca et al. (2019) focused on the biodiversity, as well as the arsenic-metabolizing microbiota inhabiting arsenic-rich groundwaters in the northern province of Italy. The presence of arsenite-oxidizing bacteria in the studied sites was confirmed in vivo by enrichment cultivation. Arsenite metabolism was consistent with the phylogeny of *aioA* genes retrieved in the environmental DNA, as well as with the enrichment of arsenite-oxidizing bacteria [258].

### 4.4. Does Existing SAR Technology Engage the Full Microbial Potential of Aquifers?

Organisms found in the well water did not occur at densities anywhere near those that could contribute significantly to arsenite removal. Weeks of incubation were needed for the communities to degrade arsenite compared to the daily cycling of the well, and a one-hundredfold dilution did away with this activity (as assayed in the laboratory) [108,238]. Under SAR conditions, microorganisms may need quite some time to grow or adapt properly to the changing environment. Or, due to alterations in the environment of the microorganisms caused by injecting aerated water into the aquifer, potential iron-cycling microorganisms may not increase as much as desired. The diurnal variation of oxygen tension may have been too rapid for them to adapt. Alternatively, the oscillation between completely aerobic and virtually anaerobic may be incompatible with the functioning of both aerobic, arsenite-oxidizing organisms and anaerobic, arsenite-oxidizing organisms. Only facultative aerobic, arsenite-oxidizing organisms might be able to thrive under the conditions below the SAR well. Iron-oxidizing *Gallionella* is a typical microaerophilic, oxygen gradient organism and should not be expected to survive under such conditions and, indeed, it was observed in the reference well but was almost absent from the SAR well [238]. A similar observation was reported for other SAR wells [89,239]. Miller (2008) also reported scant evidence for iron oxidation being dominated by microbial communities in subsurface arsenic removal field trials. Most probably, iron-oxidizing *Gallionella* requires a specific habitat [259] that was not attained in the SAR experiments of Miller (2008) or in [108,109,235,238]. Hassan (2016) [238] did not find any obvious positive effect of SAR operation on the amount of cultivable chemolithoautotrophic and heterotrophic arsenite-oxidizing bacteria in the water samples comparing post-treatment (cycle 55) to

pre-treatment. With respect to sediment, bacteria capable of growth appeared only in the post SAR sediment sample (at cycle 55) but their numbers were close to negligible (<5 CFU/mL). The numbers of heterotrophic, potentially arsenite-oxidizing bacteria were a bit more substantial in both the SAR and the reference well water. In the sediment (close to the bottom of the well), this number was even larger and increased by a factor of one thousand with SAR operation. However, one cannot be sure that under these heterotrophic conditions, all these bacterial cells engaged in arsenite oxidation, as they do not require this for their energetics. Some or all of them might be arsenite-resistant through other mechanisms than arsenite oxidation. In the laboratory experiments [108,238], they may have grown on the yeast extract provided as a carbon source. No significant variation in iron-oxidizing bacterial growth between pre and post-treatment water samples was observed either [238]. The 16S rRNA amplicons sequencing data suggested that aerobic, arsenite-oxidizing bacterial species were and continued to be rare. Potential evidence was found only for the presence of three known aerobic, arsenite-oxidizing bacterial species (*Bosea*, *Rhizobium*, and *Bradyrhizobium*). Anaerobic, arsenite-oxidizing *Dechloromonas* sp. was also rare, at only 0.5% [238]. Nitrate-reducing, iron-oxidizing species were 10 times higher (0.1%) than aerobic-oxidizing groups (*Acidovorax*, *Paracoccus*, and *Aquabacterium* spp.,). These findings suggest that the microbes of potential interest to SAR could not maintain themselves much during SAR treatment.

*4.5. bSAR: Strategies Aimed at Promoting Microbial Contributions to SAR*

In the previous section, we discussed that the microbial population in the SAR wells did not correspond with what should be optimal for SAR. Iron reducers rather than iron oxidizers were amplified. We will call any strategy optimizing microorganism activity in the well (or tank) "biotic SAR" or "bSAR", as opposed to the abiotic SAR (aSAR), where systems microbiology was not part of the strategy. Such strategies may go as far as injecting microorganisms of desired performance, pre-grown perhaps in a rooftop water reservoir, into the well, in a procedure called "bioaugmentation". They may also just try to improve the well conditions in such a manner that the more useful microorganisms thrive at the cost of microorganisms that would detract from the desired SAR performance. The latter procedure is called bio-remediation [260]. There are indications that bSAR could be effective. Katsoyiannis and Zouboulis (2004), for instance, showed that both chemolithotrophic *Gallionella* and heterotrophic *Leptothrix ochracea* were capable of oxidizing iron, and possibly arsenite, in a fixed bed filtration unit treating arsenic-contaminated groundwater, with a 95% removal efficiency of arsenite [49].

*4.6. How Could bSAR Be Enhanced: Bioremediation versus Bioaugmentation?*

In bioaugmentation, both the oscillation between aerobic and anaerobic conditions and the change with time of the iron and arsenic concentrations would cause problems for the injected organisms too. In addition, the aquifer chemistry and microbiology tend to be heterogeneous [107,238]. Highly specialized microorganisms may not be capable of adapting to all the sites' settings. Although bioaugmentation may appear to be a perfect solution to contaminated soil [261], it can have drawbacks. For example, the wrong type of bacteria can result in potentially clogged aquifers or the remediation may be incomplete or unsatisfactory. The alternative of artificial recharge with inorganic chemicals or organic carbon sources to stimulate the growth of endogenous microorganisms through bioremediation may also cause secondary pollution to the subsurface, however.

The fact that with every 24 h cycle, 3 m$^3$ of water may be run through the SAR well, is itself a strong impediment for in situ enrichment and, hence, for bioremediation: assuming that the flow is essentially convective, any amplification of organisms resident in the relevant water volume below the well is annihilated every cycle, unless the microorganisms cling to fixed soil. Also, organisms that adhere to the surface of the SAR tank (Figure 4) constitute an exception to this: they may be enriched during subsequent cycles, and perhaps this accounts for the substantial ammonia oxidation observed in the tank when samples

at the end of the aeration phase were analyzed. This enrichment might be enhanced by inserting extra surfaces into the tank.

Bioremediation is more suitable for soil with a low level of contaminants, whilst the engineered bioaugmentation method may work better in highly contaminated areas [262]. Under laboratory conditions, iron and arsenic can be oxidized and reduced biologically within a couple of days, which is, however, longer than the turnaround time in SAR as it exists. Indigenous and engineered microorganisms can provide good options both for in situ and ex situ arsenic removal technologies. Such microorganisms come with limitations, however. Sometimes, they take considerable time for adaptation to the relevant environment if environmental factors such as temperature, pH, substrate concentrations, and $O_2$ tension do not correspond to what is optimal for their growth. In addition, the bio-augmenting microbes would have to compete with the indigenous bacterial microflora, and this competition could become tough if the native microbial cell number exceeded $10^8$–$10^9$ per gram of soil (approximately 1% $w/w$) sediment.

### 4.7. How Can bSAR Best Be Enhanced?

Intrinsic in situ bioremediation may often be a slow process due to slowly growing and adapting microorganisms, limited availability of electron acceptors and nutrients, low temperatures, and high concentrations of toxic contaminants [263]. When site conditions are not suitable, bioremediation requires the construction of engineered systems. Nets of materials selected for their attraction in microbial growth may be used. Such engineered in situ bioremediation should then accelerate the desired biodegradation reactions by encouraging the growth of more microorganisms via optimizing physicochemical conditions [264]. The assemblage of growing microbial biofilms on disc-like supportive objects inserted into the SAR tank along with PVC tubing, or the insertion of soil particles and sieves, may be a good strategy for reducing perturbation during the abstraction and injection of water because there should then be less chance to wash out the microbes. Microbes need a proper time of incubation for their growth. Oxygen and other electron acceptors (e.g., $NO_3^-$ and $SO_4^{2-}$) and nutrients (e.g., phosphate) may promote appropriate microbial growth on such surfaces. When the contamination that needs to be remedied is deeper, amended water should be injected through wells. In some in situ bioremediation systems, extraction and injection are used in combination in order to control the flow of contaminated groundwater for it to be combined with above-ground bioreactor treatment and the subsequent reinjection of a nutrient-spiked effluent [265].

Engineered bSAR could further benefit from advanced bio-sparging techniques [266]. Bio-sparging is an in situ technique that uses indigenous microorganisms to remedy contamination at or below the water table boundary. It involves injecting air (or oxygen) and nutrients (in gaseous form) into the saturated zone to boost the biological activity of the local microorganisms. Air may be introduced via pipes sunk into the contaminated area and may then form bubbles in the groundwater. The extra oxygen made available in this way dissolves into the water, increasing the aeration of the overlying soil and thereby stimulating the activity of resident facultative aerobic microbes and speeding up their natural ability to metabolize the polluting substances. A number of contaminants have been successfully addressed with bio-sparging technology, including gasoline components such as benzene, toluene, ethylbenzene, and xylenes (BTEX) and other semi-volatile organic compounds (SVOCs) [267].

bSAR can be operated at the field level by measuring and calculating the number/densities and types of microorganisms present in the aquifer, followed by examining the availability of nutrients (e.g., carbon, nitrogen, and phosphorus), pH, temperature, $O_2$, Fe, and concentration of pollutants. An approximation of minimum nutrient requirements can be based on the stoichiometry of the overall biomass synthesis process [266]:

$$\text{C-source} + \text{N-source} + O_2 + \text{Minerals} + \text{Nutrients} \rightarrow$$
$$\text{Cell mass} + CO_2 + H_2O + \text{other metabolic by-products} \tag{11}$$

Different empirical formulas of bacterial cell mass have been proposed; the most widely accepted are $C_5H_5O_2N$ and $C_{60}H_{87}O_{32}N_{12}P$. Using the empirical formulas for cell biomass and other assumptions, the carbon: nitrogen: phosphorus ratios necessary to enhance biodegradation fall in the range of 100:10:1 to 100:1:0.5, depending on the constituents and bacteria involved in the biodegradation process [266]. When the actual growth substrates are known, flux balance analysis (FBA) can establish the optimal ratios of the availabilities of the various nutrients [268]. Using these stoichiometric ratios, the need for nutrient addition can be determined by using the average concentration of the constituents (carbon source) in the soils to be treated. If nitrogen addition is necessary, slow release sources should be used. Nitrogen addition can alter the pH, depending on the amount and type of nitrogen added, something that can again be calculated by using FBA (flux balance analysis). Dissolved ferrous iron [Fe(II)] in groundwater can reduce the permeability of the saturated zone soils during sparging operations as precipitating iron oxides may cause clogging. Bio-sparging may be effective if the dissolved ferrous iron concentration is < 10 mg/L [266]; otherwise, abiotic iron oxidation may swamp the corresponding microbiological processes. Slow aeration might be beneficial so as to maintain microaerobic conditions. As every location has its own microbial community structure, genomic diversity, and hydrochemistry, the remediation required for successful SAR may vary from location to location.

*4.8. Outlook: How Should bSAR Be Developed Further?*

Systems biology (Figure 9) should enable the researcher to explore the complex networks at the molecular, cellular (catabolic activity), population, microbial community (endogenous species composition), and ecosystem levels [269,270], notably by integrating precise experimental information with what is already stored in databases as well as with physical, chemical, and biological principles. It does this through both physiological and meta-genome wide experimentation and mathematical modeling.

The complex networks around bioremediation used to be approached by "black box" engineering [271]. Since the genomics–systems biology revolution, abilities to measure and model the functional microbial community structure and its stress responses in the environment at all levels have increased tremendously. Importantly, genomics has become so specific that black-boxing is no longer needed; explicit models have become possible. Bioremediation is a case of multiscale complexity that is not amenable to the traditional reductionist approaches (e.g., one compound, one strain, and one pathway) that have dominated many studies on biodegradation. To get started, one should navigate the various layers of complexity that separate the occurrence of distinct gene clusters encoding catalytic activities in single genomes all the way to extensive implementation of such a catalysis on a target site (Figure 10) [270].

The bSAR concept should also cover the multiscale complexity involved in the removal of toxic arsenic from polluted sites. Metabolic activities in the environment should be identified for the biodegradation of any given substrate [S] through a multistep biochemical route $S \rightarrow CO_2 + H_2O$. This route may require the action of a single performer microorganism, endowed with all enzymes required for the complete mineralization of the compound or by a line-up of microorganisms each catalyzing only part of the entire route, yet able to benefit from it in terms of Gibbs energy extraction or otherwise. A number of processes *upstream* (diffusion in solid matrixes, bioavailability, weathering, and abiotic catalysis of pollutants [272]) and *downstream* (stress, predation, and competition [273]) of the biochemical route will constrain the outcome of the whole action. Peripheral biodegradation routes need to be coupled to the central metabolism and to the overall Gibbs energy transduction of the cells. Biodegradation should be linked to growth or detoxification in order to provide a selective advantage to the cells that bear the catalytic activity [274]. But, unlike the chemical and biochemical aspects where approaches such as flux balance analysis may help, such microbial growth facets of biodegradation are more difficult to implement in a predictive system, although, here, the new "dynamic competition FBA" may

help [275]. Microbial communities contain multiple variants of *pan enzyme* (corresponding to enzymatic activity without cell borders) [270,276] with non-identical efficiencies [277]. These further complicate the analyses.

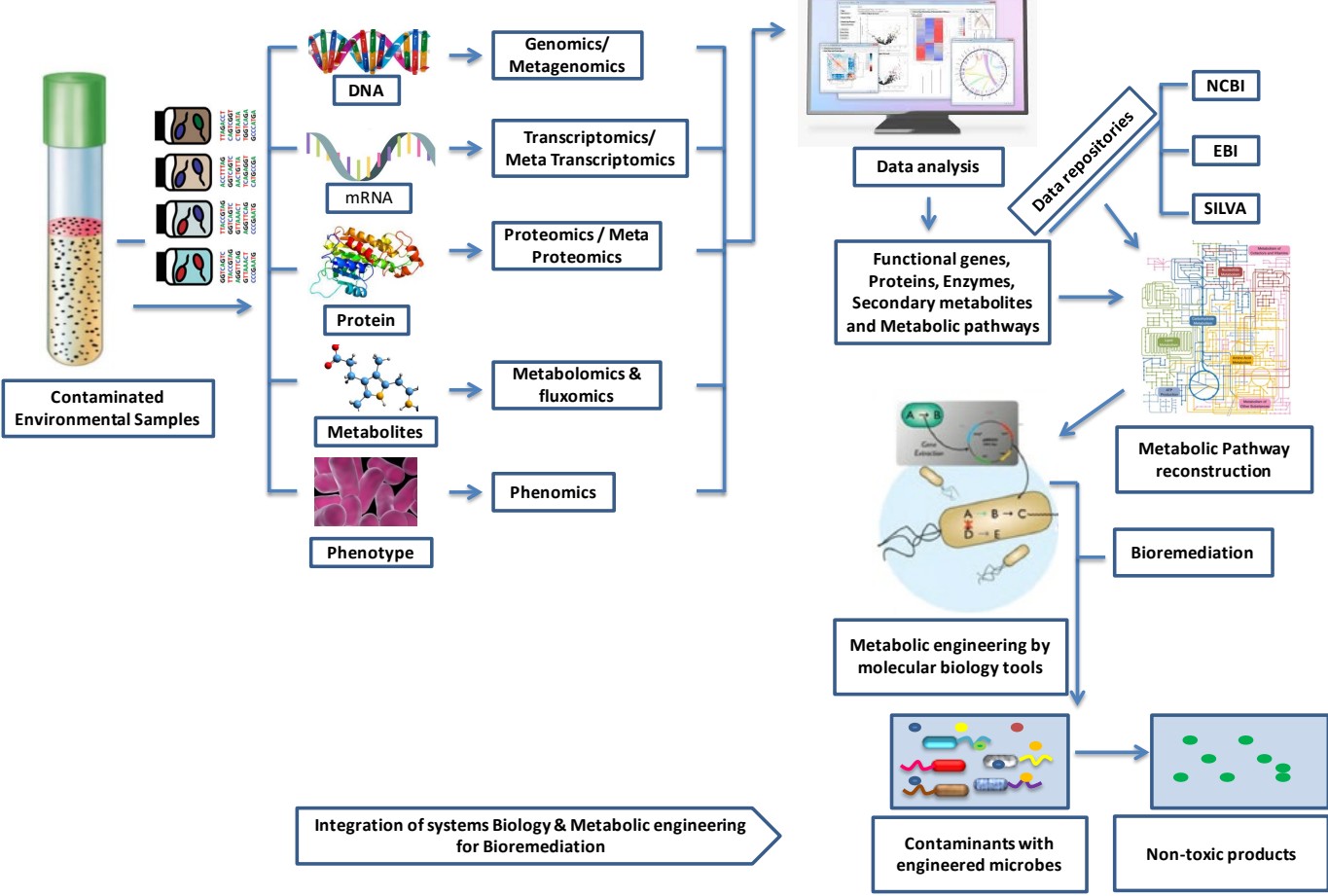

**Figure 9.** The integrated and iterative metagenome-wide approach of systems biology. Systems biology provides valuable detailed information about biological processes. Then, bioremediation and bioaugmentation can be used to optimize soil and ground water.

Biological systems maintain phenotypic stability in the face of diverse perturbations imposed by the environment, stochastic events, and genetic variation [278]. Experiments under laboratory conditions that mimic the variation of the conditions in situ for every potential SAR well will be necessary in order to obtain a validated predictive understanding of the functioning of microbes and their geochemical interactions in the context of a particular well's SAR. Thus, systems microbiology individualized for each drinking water well, is necessary. This should also help us understanding the potential removal mechanisms and tell us about the sustainability and acceptability of SAR in field applications. Because of the complexity of bSAR, such experimentation should be precise and assisted by mathematical modeling, much as has been done for the systems chemistry of arsenite removal [98] and for intracellular systems biology at large. Each systems biology model of bSAR will be even more complex than that of systems chemistry, accommodating multiple organisms at densities and activities that vary over time and with conditions.

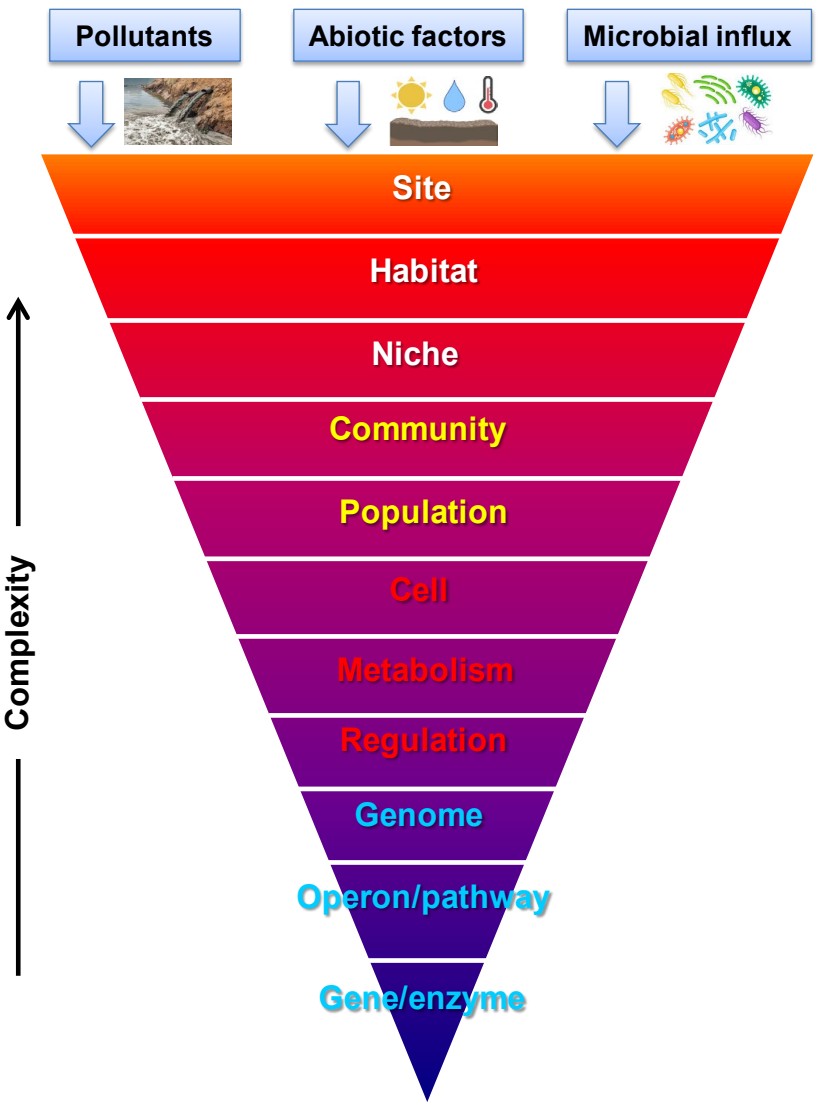

**Figure 10.** The multiscale complexity pyramid that one has to go through for taking aboard all factors that intervene in the implementation of any bioremediation strategy. Note that this is a highly dynamic situation as the course of the biocatalysis changes both the chemical profile of the pollutants and the structure of the microbial community and vice versa. Pollutants and the side products of their metabolism can also have a strong mutagenic effect on the microbial genomes and affect the architecture of the abiotic scenario (adapted and modified with permission [270]).

To obtain a high efficiency of the in situ removal of arsenic by using microbes that catalyze arsenite oxidation by nitrate or oxygen, optimal microbial growth is also required. A simple kinetic (Monod equation) microbial model may be applied to the bioremediation of arsenic both in situ and in column experiments (ex situ). This model should also inform us about the microbial growth process and the optimal number of cells required for arsenic removal performance. Several studies have developed empirical models [24,279,280] that describe microbial dynamics by quantifying microbial growth and decay for specific arsenite oxidizers. Microbial growth rates were assumed to depend on the availability of arsenite [As(III)] as an electron donor and nitrate as an electron acceptor, with the Gibbs energy derived being used to fix carbon into organic material and maintain the cell according to the slightly modified equations of Wallis et al. (2010) [280]:

$$6\,NO_3^- + 5\,HCO_3^- + 19\,HAsO_3^- \rightarrow C_5H_7O_2N + 5\,NO_2^- + 2\,H_2O + 19\,HAsO_4^{2-} + 32\,H^+ \tag{12}$$

The mass balance of the arsenite-oxidizing microbial group is as follows:

$$\frac{\partial X}{\partial t} = \left(\frac{\partial X}{\partial t}\right)_{growth} + \left(\frac{\partial X}{\partial t}\right)_{decay} \tag{13}$$

$\left(\frac{\partial X}{\partial t}\right)_{growth}$ is the rate of the above chemical reaction of biomass synthesis. Microbial growth may be simulated using a standard Monod kinetic growth model and a first-order biomass decay term:

$$\left(\frac{\partial X}{\partial t}\right)_{growth} = Y \cdot V_{\max} \cdot X \cdot \frac{C_{As(III)}}{K_{As(III)} + C_{As(III)}} \cdot \frac{C_{NO_3^-}}{K_{NO_3^-} + C_{NO_3^-}} \tag{14}$$

$$\left(\frac{\partial X}{\partial t}\right)_{decay} = -k_{decay} \cdot X \tag{15}$$

Here $X$, $C_{As(III)}$, and $C_{NO_3^-}$ represent the microbe, arsenite, and nitrate concentrations, respectively. $k_{decay}$ (with unit per hour) and $V_{\max}$ are the decay rate constant of the biomass and the maximum specific (i.e., in Mole per gram per hour) uptake rate constant of arsenite, respectively. $Y$ is the growth yield (gram per Mole of arsenite taken up) and $K_{As(III)}$ and $K_{NO_3^-}$ are the Monod constants. Mostly, either arsenite or nitrate and not both would be limiting, meaning that the concentration of the substrate that is not limiting remains far above the corresponding $K$, so that the corresponding factor in the equation may be taken to disappear. More details have been discussed elsewhere [280,281]. Using a second aspect of systems biology, i.e., flux balance analysis [268], on genome-wide metabolic maps, optimal supply rates of nutrients and the resulting production of metabolites and species of arsenic and iron can be calculated. Recently, a type of FBA that accommodates competition between different cell types for a given influx of substrate has become available [275], which should also be relevant.

It is now the challenge to upgrade these simpler modeling procedures to a more comprehensive systems model, which takes into account the change in time and space of the arsenite and nitrite concentrations, the parallel process of the aerobic oxidation of arsenite by other organisms, as well as the precipitation of arsenate and arsenite with the ferrihydrite that is formed, with the accompanying acidification being taken into account. The calibration of such a model will require more measurement of the conditions in the well and their variation with time. After measuring the parameters in these equations, one may then be able to design conditions that would enable the microorganisms to grow to sufficient densities to be able to remove most of the arsenite from SAR wells that are provided with Fe(II) and oxygen periodically, at time intervals determined by the models. In particular, the suggestions given in Section 4.2 may be tested, i.e., (i) a slow (micro-)aerobic phase followed by a quick phase of water extraction, (ii) the specific stimulation of arsenite-oxidizing microorganisms, and (iii) optimization of the volumes and origins of the water pumped out of and reinjected into the drinking water well and surrounding wells.

In parallel to such field experiments, a laboratory column setup [229] will be necessary for the development of realistic models. In addition, larger-scale laboratory mimicries of actual drinking water wells will be necessary to obtain robust predictive strategies and models for effective bSAR and to be able to pre-validate these models and strategies. Figure 11 depicts an experimental set up for a (large) field laboratory. In view of the global importance of arsenite toxicity, worldwide support for such a setup in one of the most affected countries is rational.

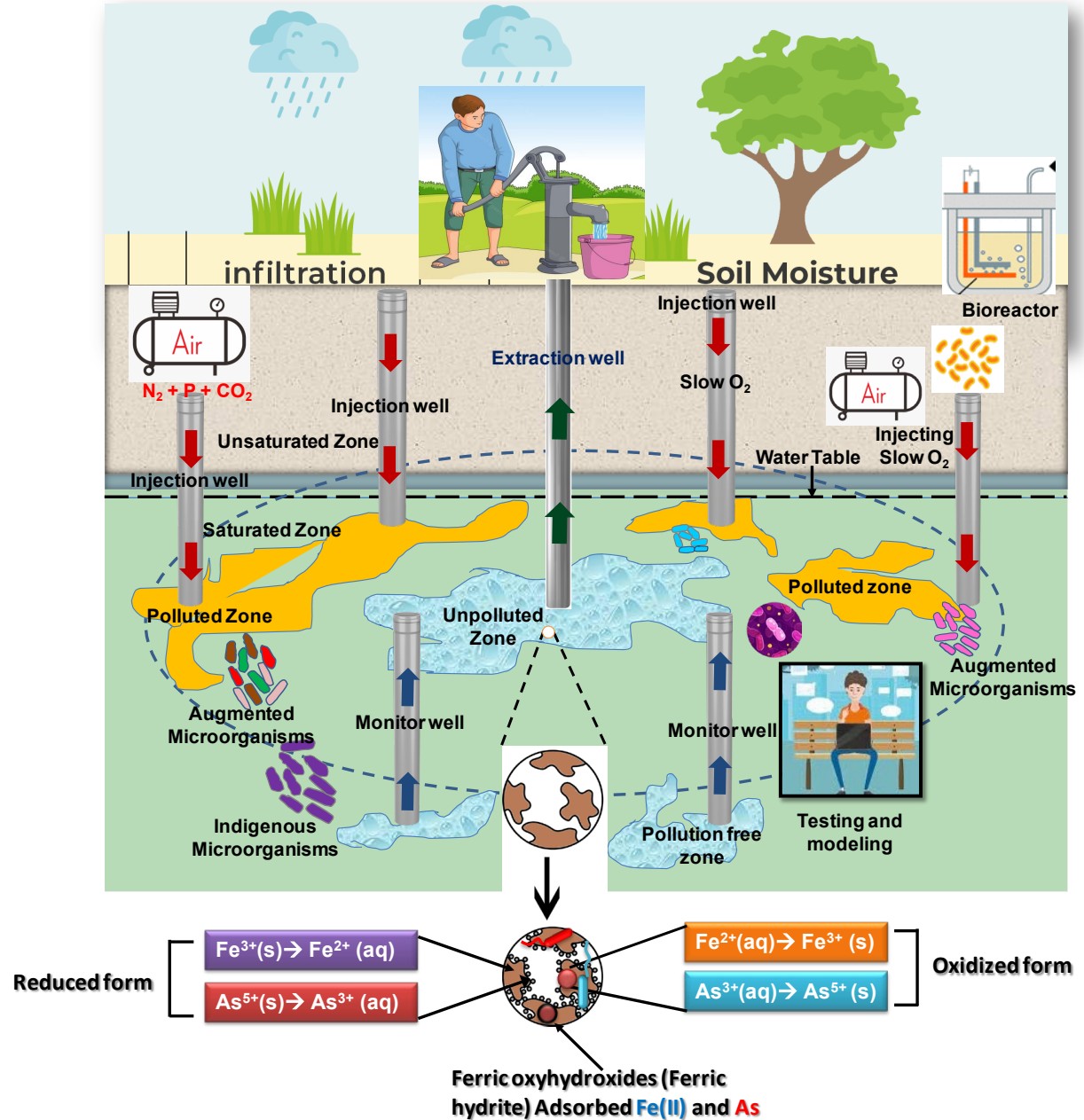

**Figure 11.** Proposed model of bSAR using engineered microorganisms for up-scaling the bioremediation of arsenic through bioaugmentation. Distinct extraction and injection wells (proposed fluid flow indicated by arrows) are used in combination in order to integrate the control of the flow of contaminated groundwater with above-ground bioreactor treatment and the subsequent reinjection of nutrient-spiked effluent. Injection wells are sunk at the periphery of the contaminated site and $O_2$ is sparged periodically, but at a slower than diurnal rhythm and at a slower rate or lower partial pressure. Monitoring wells are used to identify the contamination level, the activity of microorganisms, and the sufficient amount of nutrients available for microbial growth. At some point, if indigenous microbial load or number is reduced due to perturbation, then injection of engineered microorganisms into the subsurface would be a good option in order to augment the bioremediation process with local inhabiting microbes. Once the bSAR model has been optimized, arsenic pollution-free waters can be abstracted through the extraction well for drinking and other purposes. Combination with physical chemical methods of treating water (e.g., Section S2.3 and Figure S8) may also be integrated.

## 5. Perspectives

Aspects of the arsenic toxicity of groundwater have been the subject of great many, often excellent, reviews already, some of which are quite recent. These reviews focused on one (or a number of closely related) aspects of the topic. Ganie et al. (2023) [282], for instance, focused on arsenic toxicity in the human, which we here only discussed briefly. Dilpazeer et al. (2023) discussed arsenic management strategies and the feasibility, cost effectiveness, and merits and demerits of the different types of arsenic removal technologies in depth, including physicochemical and biological methods [283]. They also suggested that future arsenic removal technologies should be ecofriendly and sustainable in terms of providing safe drinking water to societies, especially in less developed and developing countries. In 2023 Patel et al. discussed the many routes of exposure of the human to arsenic as well as worldwide contamination, but did not go into depth by discussing any particular location in detail [20]. Cerron-Calle et al. (2023) discussed electrified technologies for removing arsenic from drinking water [284]. Hassan et al. (2023) focused on the array of arsenic-removing techniques [48]. Fatoki et al. (2022) paid most attention to the environmental toxicity of arsenic and discussed the interaction of arsenic iron sulfide and its toxicity to mammals [68]. Monteiro de Oliviera et al. (2021) discussed the pathologies arising from arsenic toxicity [285], whilst Sing et al. (2021) focused on the spatial distribution of arsenic contamination in an aquifer in central India [286]. Shaji et al. (2021) considered the entire Indian peninsula whilst focusing on public health, human toxicity, and policies [19].

Uppal et al. (2019) only gave a brief overview of the situation in various countries, particularly in Southeast Asia [287]. Likewise, Ahmad and Bhattacharya (2019) [288], Yunus et al. (2016), and Sing and Stern (2017) presented brief but interesting overviews and alerts [289,290]. The book by Hassan et al. (2018), *Arsenic in Groundwater Poisoning and Risk Assessment*, has many chapters, but they are mainly on toxicology and public health, with little about microbiology, arsenic and iron chemistry, or the integration of all these aspects [291]. Bhowmick et al. (2018) had an interesting focus on biomarkers of arsenic toxicity [292].

All these aspects are important for the arsenic toxicity of groundwater and all these reviews are worth studying, therefore. In this review, we have seen, however, that arsenic toxicity is not determined by the mere sum of all the effects discussed in these reviews: the various effects amplify and/or ameliorate each other through interactions often depending on external conditions such as pH, redox potential, and the presence of substances accelerating microbial growth. It is in highlighting these cross-influences that the various processes discussed individually in the above reviews have on each other, that the present review adds to the already existing ones: in the present review, we have focused on how arsenic toxicity is the net effect of a substantial number of processes that interact nonlinearly. The interactions between the processes make analysis and outcome prediction complex and, indeed, we have discussed how the outcome of existing attempts at subsurface arsenic removal (SAR) have been disappointing. Examples of these nonlinearities were the co-precipitation of arsenate with ferric iron as well as the effect of the oxidation of ferric iron on the oxygen tension and thereby on the density of aerobic microorganisms playing key roles in oxidizing arsenite or ferrous iron. The ability of microbes to enhance their density greatly when they can extract Gibbs energy for growth from their environment constituted another set of examples. These may relate to the organisms' use of arsenite or ferrous iron as electron donors or to their consumption of arsenate or ferric iron as electron acceptors when organic chemicals in the environment allow for heterotrophic growth. The oxygenation and reinjection of groundwater thereby has complex effects on arsenic mobility: whether it oxidizes the arsenite and immobilizes it as arsenate depends on whether sufficient ferrous iron is oxidized to ferric iron oxide and precipitated as such. The presence of various microbial species, as evidenceable by pangenomic analyses, may enhance the oxidation of ferrous iron considerably, but may also remobilize arsenic by the reduction of the arsenate or ferric iron once the oxygen has been consumed.

The qualitative state and dynamics of nonlinear systems tends to depend on the values of their parameters, and one of the perspectives that this review offers is that of detailed experimental analyses (see Figure 11) of multiple and different sites of arsenic pollution. A second perspective is that of system-wide collection of multiple data sets, such as pangenomic analyses of the metabolic and growth capabilities of all microbial species present at a site. A third perspective is that of including the measurement of important thermodynamic parameters such as pH and ambient redox potential (and dissolved oxygen concentrations) and an assessment of their effects on the relative abundance of the various forms of arsenic and iron. We have illustrated this in the equilibrium thermodynamic sense (e.g., Figure 2) but, ultimately, time-dependent measurements will be necessary, as some processes appear to be slow (see Section 4.4). When processes and their cross-connections are nonlinear, it becomes impossible to understand and predict the behavior of a system by intuition or interpolation/extrapolation. One therefore needs to invoke the assistance of mathematical modeling and chemical and biological theory, which is our fourth perspective.

The present review has only been able to highlight some of the more important nonlinear interactions. It has not been able to review much of the modeling, for the simple reason that the modeling of the nonlinear interactions determining the arsenic toxicity of groundwater is still in its infancy. A pessimist might even suggest that any aim towards an integral understanding of the arsenic toxicity of groundwater is too ambitious. We are more optimistic because of a number of reasons: (i) the genomics revolution has dramatically enhanced the ability to determine the microbial population of a groundwater site, (ii) the proteomics and metabolomics revolutions will soon enable a further identification at the level of proteins and metabolites, (iii) progress in bioinformatics and systems biology of single species has recently extended to ecosystems as well as to flux predictions therein, and (iv) the software and capabilities of mathematical modeling have also increased sharply. Our fifth, more integral, perspective, therefore, is the development of a systems biology of the arsenic toxicity of groundwater.

**Supplementary Materials:** The supplementary material includes Figures that are cited in the main text, extra supplementary information in text form, as well as relevant references [293–365]. The following supporting information can be downloaded at https://www.mdpi.com/article/10.3390/toxics12010089/s1: Figure S1: Estimated numbers of people (in billions) in various parts of the world with unsafe drinking water (online data were obtained from WHO/UNICEF as of 2020). Figure S2. Negative redox potential–acidity ($-E/pH$) diagram (Pourbaix diagram) indicating the thermodynamically lowest Gibbs energy form of arsenic at 25 °C and 101.3 kPa total pressure (1 atmosphere) at physical–chemical standard conditions (1 M, 1 Bar). Figure S3. Uncoupling of substrate-level phosphorylation in glycolysis by arsenate. Figure S4. Negative redox potential versus acidity ($-E$ versus pH) diagram indicating the thermodynamically lowest Gibbs energy form of iron at 25 °C and 101.3 kPa total pressure (1 atmosphere) at relevant standard conditions (21% $O_2$, 0.55 μBar $H_2$, 10 μM ferrous and ferric iron). Figure S5. Simplified 2-dimensionalized structure for goethite, inclusive of the surface (on the right hand side). Figure S6. Combined negative redox potential ($-E$) versus pH diagram for iron and arsenic for groundwater reference conditions (21% $O_2$, 0.55 μbar $H_2$, 10 μM ferrous and ferric iron, 10 nM arsenates and arsenites, 1 mbar arsine). Figure S7. Putative electron transport chain for heterotrophic arsenite oxidation in Hydrogenophaga NT14, which can extract Gibbs energy from arsenic oxidation by molecular oxygen. Figure S8. Principle of the solar method with illumination, photochemical formation from $O_2$ of reactive oxidants for the oxidation of As(III) to As(V), and precipitation of Fe(III) (hydr)oxides with adsorbed As(V). Supplementary text.

**Author Contributions:** Conceptualization, Z.H. and H.V.W.; methodology, Z.H.; formal analysis, Z.H. and H.V.W.; investigation, Z.H. and H.V.W.; writing—original draft preparation, Z.H.; writing—review and editing, H.V.W.; visualization, H.V.W.; supervision, H.V.W.; project administration, Z.H. and H.V.W. All authors have read and agreed to the published version of the manuscript.

**Funding:** This research received no external funding.

**Institutional Review Board Statement:** Not applicable.

**Informed Consent Statement:** Not applicable.

**Data Availability Statement:** No new data were created.

**Acknowledgments:** We are much indebted to the late Wilfred Röling for his invaluable help and advice, which made this project possible.

**Conflicts of Interest:** The authors declare no conflicts of interest.

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
