# Peer review of "Arsenic Contamination of Groundwater Is Determined by Complex Interactions between Various Chemical and Biological Processes"

_toxics, doi:10.3390/toxics12010089_

Round 1

Reviewer 1 Report

Comments and Suggestions for Authors

The authors have presented interesting information about As, its behavior, microbiology, and toxicity. In my opinion, there is a lot of information that is not necessary to include since the manuscript has as its research object a SAR (Subsurface Arsenic Removal) system.

The information that does not need to be included could be another theoretical manuscript.

I would recommend that authors exclude headings from 1 to 2.6, you can also exclude heading 3.

The authors must include an introduction in which the background on the topic (SAR) is presented, and the objective of their research (we discuss how consideration of the system's biological aspects of SAR may lead to new avenues towards safer drinking water, by creating conditions that optimize groundwater microbial ecology vis-à-vis reduced arsenic toxicity).

Reviewer 2 Report

Comments and Suggestions for Authors

The manuscript Systems Approaches to Arsenic Contamination of Drinking Water.

This is a review paper generally well written., which is important because arsenic is a contaminant posing a serious threat to public health in many places in the world. The authors presented papers dealing with the arsenic pollution level in different countries around the world, presented in detail arsenic chemistry, its speciation, interaction with ferrous and ferric iron and toxicity of its inorganic and organic species in the environment.

Also, the authors well reviewed papers presenting the remediation strategies of arsenic removal.

A section of biology influence on arsenic concentration and speciation in water is presented (influences of organic matter, microorganisms). All the findings are well illustrated in 11 Figures in the main manuscript and 8 figures in Supplementary Materials.

The manuscript may be improved by including several up-to-date references, since many of the cited literature is older that 10-15 years. In the last years there are many woks dealing with As occurrence in water (e.g. doi.org/10.1016/j.kjs.2022.12.001; doi.org/10.1016/j.ecoenv.2023.114880; doi.org/10.1016/j.chemosphere.2016.12.130; doi.org/10.1155/2017/3037651)

Reviewer 3 Report

Comments and Suggestions for Authors

The authors conducted a review titled "System approaches to arsenic contamination of drinking water." First, the content mainly focused on case studies in Bangladesh. Second, the terms used in the title are ambiguous and not clearly defined. For example, what exactly are the "system approaches"? The full name of "SAR" should be provided when it first appears. Also, "drinking water" refers to groundwater in Bangladesh in the global context. Hence, this can be misleading. Third, the introduction of iron chemistry when the topic is on arsenic opens the scope too wide. Fourth, the biochemical treatment mentioned can be more concise. If the work is on biochemical treatment, it would be better to rephrase the title to better suit its content. Fifth, the authors failed to highlight the rationale for this review - why do the authors want to do this review? How are case studies in Bangladesh relevant to other countries?

Round 2

Reviewer 3 Report

Comments and Suggestions for Authors

There are already many review papers on arsenic contamination in groundwater, the use of ambiguous terms like "system approaches" does not help at all. In terms of mechanisms that lead to arsenic contamination or solutions that mitigate the contamination, they have already been covered in review papers as well. The authors need to redefine the scope of this review paper to make this work different from other review papers. This work focused heavily on the water chemistry of As and Fe in the first half but failed to link it to the latter half on microbiology. How is 2.1.1 related to chemical kinetics? no rates were mentioned. Same for 2.1.2 and 2.1.3 - I do not think the use of "kinetics" is appropriate here.

I would suggest the authors either focus on the biochemical mechanisms leading to contamination or bioremediation, not both at the same time.

Round 3

Reviewer 3 Report

Comments and Suggestions for Authors

The authors have made substantial efforts to address the issues mentioned.